

# The Fully Coupled Regionally Refined Model of E3SM Version 2: Overview of the Atmosphere, Land, and River

Qi Tang[1], Jean-Christophe Golaz[1], Luke P. Van Roekel[2], Mark A. Taylor[3], Wuyin Lin[4], Benjamin R. Hillman[3], Paul A. Ullrich[5], Andrew M. Bradley[3], Oksana Guba[3], Jonathan D. Wolfe[2], Tian Zhou[6], Kai Zhang[6], Xue Zheng[1], Yunyan Zhang[1], Meng Zhang[1], Mingxuan Wu[6], Hailong Wang[6], Cheng Tao[1], Balwinder Singh[6], Alan M. Rhoades[7], Yi Qin[1], Hong-Yi Li[8], Yan Feng[9], Yuying Zhang[1], Chengzhu Zhang[1], Charles S. Zender[10], Shaocheng Xie[1], Erika L. Roesler[3], Andrew F. Roberts[2], Azamat Mametjanov[9], Mathew E. Maltrud[2], Noel D. Keen[7], Robert L. Jacob[9], Christiane Jablonowski[11], Owen K. Hughes[11], Ryan M. Forsyth[1], Alan V. Di Vittorio[7], Peter M. Caldwell[1], Gautam Bisht[6], Renata B. McCoy[1], L. Ruby Leung[6], and David C. Bader[1]

[1]Lawrence Livermore National Laboratory, Livermore, CA, USA
[2]Los Alamos National Laboratory, Los Alamos, NM, USA
[3]Sandia National Laboratories, Albuquerque, NM, USA
[4]Brookhaven National Laboratory, Upton, NY, USA
[5]Department of Land, Air and Water Resources, University of California, Davis, CA, USA
[6]Pacific Northwest National Laboratory, Richland, WA, USA
[7]Lawrence Berkeley National Laboratory, Berkeley, CA, USA
[8]Department of Civil and Environmental Engineering, University of Houston, TX, USA
[9]Argonne National Laboratory, Lemont, IL, USA
[10]Departments of Earth System Science and Computer Science, University of California, Irvine, CA, USA
[11]Department of Climate and Space Sciences and Engineering, University of Michigan, Ann Arbor, MI, USA

**Correspondence:** Qi Tang (tang30@llnl.gov)

**Abstract.** This paper provides an overview of the United States (US) Department of Energy's (DOE's) Energy Exascale Earth System Model version 2 (E3SMv2) fully coupled Regionally Refined Model (RRM) and documents the overall atmosphere, land, and river results from the Coupled Model Intercomparison Project 6 (CMIP6) DECK (Diagnosis, Evaluation, and Characterization of Klima) and historical simulations – a first-of-kind set of climate production simulations using RRM. The North American (NA) RRM (NARRM) is developed as the high-resolution configuration of E3SMv2 with the primary goal of more explicitly addressing DOE's mission needs regarding impacts to the US energy sector facing Earth system changes. The NARRM features finer horizontal resolution grids centered over NA, consisting of 25→100 km atmosphere and land, 0.125° river routing model, and 14→60 km ocean and sea ice. By design, the computational cost of NARRM is ∼3x of the uniform low-resolution (LR) model at 100 km but only ∼10-20% of a globally uniform high-resolution model at 25 km.

A novel hybrid timestep strategy for the atmosphere is key for NARRM to achieve improved climate simulation fidelity within the high-resolution patch without sacrificing the overall global performance. The global climate, including climatology, time series, sensitivity, and feedback, is confirmed to be largely identical between NARRM and LR as quantified with typical climate metrics. Over the refined NA area, NARRM is generally superior to LR, including for precipitation and clouds over the contiguous US (CONUS), summertime marine stratocumulus clouds off the coast of California, liquid and ice phase clouds



near the North polar region, extratropical cyclones, and spatial variability in land hydrological processes. The improvements over land are related to the better resolved topography in NARRM, whereas those over ocean are attributable to the improved air-sea interactions with finer grids for both atmosphere and ocean/sea ice. Some features appear insensitive to the resolution change analyzed here, for instance the diurnal propagation of organized mesoscale convective systems over CONUS, and the warm-season land-atmosphere coupling at the Southern Great Plains. In summary, our study presents a realistically efficient

approach to leverage the RRM framework for a standard Earth system model release and high-resolution climate production simulations.

# 1    Introduction

Global Earth system models (ESMs) are fundamental tools for understanding the past evolution of the climate system and projecting future climate changes under various anthropogenic scenarios. High horizontal resolution simulations on climate scales

have been recognized as one of the increasingly important directions of ESM development in recent years (Demory et al., 2014; Haarsma et al., 2016). Compared to low-resolution models, high-resolution models show superior fidelity in representing both the large-scale circulation (e.g., meridional ocean heat transport) (Griffies et al., 2015) and small-scale processes (e.g., clouds and streamflow) (Haarsma et al., 2016, and references therein). More importantly, simulations with enhanced horizontal resolution exhibit improved skills in capturing regional climate change signals and facilitating process-level studies, which provide

a crucial basis for assessing the impacts of climate extremes with augmented societal implications. However, fine resolution and multi-century simulations (with ensembles) are competing requirements for climate experiments due to limited computational and human resources. This conflict will likely continue to challenge the climate modeling community as evidenced by the fact that more than three times (72 vs. 23) as many model sources (including different versions of the same model) have published simulations at 100-km than at 25-km nominal resolutions in the current Coupled Model Intercomparison Project 6

(CMIP6) archive (https://esgf-node.llnl.gov/search/cmip6/, access date: 18/08/2022). This suggests that despite the commonly recognized benefits, not many modeling centers can afford to pursue routine high-resolution climate simulations.

The Energy Exascale Earth System Model (E3SM) project (Leung et al., 2020) is supported by the U.S. Department of Energy (DOE) with a primary goal of improving actionable predictions of Earth system variability and change by leveraging advanced DOE computational resources. Scientifically, E3SM development is motivated by modeling requirements in three

overarching fields (i.e., water cycle, biogeochemistry, and cryosphere) to address the most critical DOE mission-related questions, such as water availability, wildfires, heat waves, and sea-level rise, which all pose challenges to the energy sector with climate change. High-resolution simulations are clearly more desirable to achieve these E3SM objectives since these processes have high spatiotemporal variability. However, uniformly increasing the grid size for climate production simulations is not an easy task even with DOE's world class high performance computing power. For example, the 25-km simulation is at least 32

times (16x more grid cells, 2x smaller physics timestep, and 4x smaller dynamical core timestep) more expensive than the 100-km version with the E3SM version 1 (E3SMv1) model (Caldwell et al., 2019), making high-resolution models not only much more computationally expensive to run but also to tune for skillful simulations. With these demands and limitations, a





multiscale approach is an attractive avenue for global ESMs to deliver high-resolution production simulations over target areas at a more economical cost.

The multiresolution method (Ringler et al., 2008; Leung et al., 2013), also known as regionally refined model (RRM) or variable-resolution (VR) model, was proposed to alleviate the computational burden of global ESMs by refining a fraction of the globe with higher resolution while keeping (without coarsening) the remaining area at lower resolution. The RRM method is a general tool for all major ESM components, such as atmosphere, land, ocean, and sea ice. With a careful design of the RRM mesh, the high-resolution grids can better represent fine-scale processes over an area of interest at a typical cost of only

~10–20% of a comparable globally uniform high-resolution configuration. Compared to regional or nested climate models, global RRMs by design minimize the impacts from the lack of a two-way dynamical feedback between the refined area and the outside domain.

Recently, an increasing number of studies have successfully applied the RRM technique in global ESMs to tackle a wide range of climate research themes from climatological statistics of idealized aquaplanet (Zarzycki et al., 2014) and mean climate

state of more realistic simulations (Sakaguchi et al., 2015, 2016; Gettelman et al., 2018; Tang et al., 2019) to complex terrain climate (Wu et al., 2017; Rhoades et al., 2018c; Rahimi et al., 2019; Bambach et al., 2021) and climate extremes (Huang and Ullrich, 2017; Rhoades et al., 2020a, b; Zarzycki et al., 2021; Reed et al., 2022; Xu et al., 2022). Others leveraged RRM to study specific aspects of climate, such as tropical cyclones (Zarzycki and Jablonowski, 2014, 2015; Hazelton et al., 2018), marine stratocumulus (Bogenschutz et al., 2022), snowpack (Rhoades et al., 2016, 2017), surface energy flux (Burakowski

et al., 2019), Greenland surface mass balance (van Kampenhout et al., 2019), irrigation impacts on regional climate (Huang and Ullrich, 2016), and land use and land cover change influence on land-atmosphere coupling and precipitation (Devanand et al., 2020). Lately, the RRM resolution has been pushed to a new limit for watershed-scale hydrology analysis (Xu and Di Vittorio, 2021) and cloud-resolving scale climate simulation (Liu et al., 2022).

The RRM high-resolution results are robust regardless of where the fine-grid patch is located, covering almost all typical

climate regimes such as the contiguous US (CONUS) (Gettelman et al., 2018; Tang et al., 2019), the western (Rhoades et al., 2016; Huang et al., 2016; Huang and Ullrich, 2017; Rhoades et al., 2018c) and eastern US (Liu et al., 2022), South America (Sakaguchi et al., 2015, 2016; Bambach et al., 2021), Asia (Sakaguchi et al., 2016), East Asia (Liang et al., 2021), eastern China (Xu et al., 2021), the Tibetan Plateau (Rahimi et al., 2019), the Maritime Continent (Harris and Lin, 2014), Atlantic basin (Zarzycki et al., 2015), Southeast Pacific (Bogenschutz et al., 2022), Greenland (van Kampenhout et al., 2019), and

Arctic (Veneziani et al., 2022). Furthermore, the RRM capability in representing the high-resolution climate seems general for different models, including Variable-Resolution Community Earth System Model (VR-CESM) (e.g., Gettelman et al., 2018), E3SMv1 atmospheric model (EAMv1) (Tang et al., 2019), Model for Prediction Across Scales-Atmosphere (MPAS-A) (Hagos et al., 2013; Sakaguchi et al., 2015, 2016; Liang et al., 2021), the Geophysical Fluid Dynamics Laboratory finite-volume dynamical core on the cubed-sphere grid (Harris and Lin, 2013, 2014), and ICOsahedral Non-hydrostatic Earth System Model

(ICON-ESM) (Jungclaus et al., 2022).

All of the aforementioned studies utilize RRMs for Atmospheric Model Intercomparison Project (AMIP-type) (Gates et al., 1999) simulations. Although these studies provide valuable experience and important knowledge about RRMs, modeling cen-




ters still face the question of how to transform such AMIP-type RRM achievements from individual scientific studies empha-
sizing specific climate aspects to a standard global ESM release version aiming at a much broader and general scope. At a

minimum, two criteria should be satisfied for the RRM to be widely adopted for global ESM releases: (1) reasonable global
climate, and (2) minimal effort of retuning based upon the low-resolution counterpart. Regarding (1), most previous AMIP-
type RRM studies focus on the regional results within the refined grids without paying much attention to the outside domain.
While this might be acceptable for targeted studies, one cannot release a global model without reasonable global results which
is additionally required to address the challenge of a long (multi-century) spin-up and demonstrate top-of-atmosphere (TOA)

radiative balance in pre-industrial fully-coupled simulations. Regarding (2), some physics parameterizations (e.g., deep con-
vection) suffer from poor scale-awareness and hence require retuning as the model resolution increases (e.g., Xie et al., 2018).
This implies significant model calibration efforts that may be unaffordable in addition to tuning the low-resolution model.

In the E3SMv1 atmosphere and land models, we demonstrated that the RRM (25→100 km horizontal resolution with the 25-
km mesh over the CONUS) can mimic the climate simulated by the uniformly high-resolution (25-km) model over the refined

mesh with substantially less computational cost (Tang et al., 2019). In the present study, starting from the lower resolution
configuration of E3SMv2 (Golaz et al., 2022), we extend the RRM configuration to ocean and sea ice as a fully coupled RRM
with fine meshes centered over North America (NA). We propose an innovative RRM strategy (see details in Section 2.1) to
meet the above two criteria and for the first time to deliver production climate simulations using a fully coupled RRM.

This paper focuses on the atmosphere, land, and river components of the E3SMv2 North American RRM (NARRM), while

a companion paper (Van Roekel et al., 2022) overviews the NARRM ocean and sea ice. This paper is organized as follows.
Section 2 describes the NARRM model, key tools and tests used to create its atmospheric configuration, and our hybrid timestep
strategy for the atmospheric component. Section 3 summarizes the simulations performed in the present study and reports on
the computational cost of the NARRM historical simulation relative to its lower-resolution (LR) counterpart. Analyses of model
results start at the global scale in Section 4 and then shift to the high-resolution NA region in Section 5 for atmosphere, land,

land-atmosphere interactions, and river. Conclusions and discussions are presented in Section 6.

## 2   Model description

Except for the mesh and mesh-related settings, E3SMv2 LR and NARRM essentially have the same atmosphere, land, and
river components. They are upgraded from E3SMv1 and briefly described here. In the E3SMv2 atmosphere model (EAMv2),
the dynamical core uses the High Order Method Modeling Environment (HOMME) package (Dennis et al., 2005, 2011; Evans

et al., 2013) on the spectral element grid (Taylor and Fournier, 2010). HOMME has been updated to use a potential temperature
formulation of the equations with a more accurate pressure gradient (Taylor et al., 2020; Herrington et al., 2022) and a new
interpolation semi-Lagrangian scheme (Islet) for passive tracer transport (Bradley et al., 2022). The physics operates on a
separate finite volume grid (Hannah et al., 2021), which has 4/9 as many columns as the corresponding spectral element grid
(see Table 1) and hence runs about 2x faster than it would on the spectral element grid. The physics parameterization updates

include the Cloud Layers Unified By Binormals scheme (Golaz et al., 2002; Larson, 2017) for subgrid turbulent transport and





cloud macrophysics; the Zhang-McFarlane (ZM) deep convection scheme (Zhang and McFarlane, 1995) with a new trigger method (Xie et al., 2019); gravity wave parameterizations following Richter et al. (2010) with additional tuning (Beres et al., 2004; Richter et al., 2019); the O3v2 package (Tang et al., 2021) for the prognostic stratospheric ozone; and the four-mode version of Modal Aerosol Module (MAM4) (Liu et al., 2016; Wang et al., 2020) with an updated treatment of dust aerosol

(Feng et al., 2022). The same set of EAM physics parameters is used in the LR and NARRM simulations analyzed here. The LR grid is a quasi-uniform $1°$ cubed sphere grid with an average grid spacing of $\sim$100 km. The NARRM grid has an average grid spacing of $\sim$25 km over North America, transitioning to match the $\sim$100 km cubed-sphere grid over the rest of the globe. All simulations, except the idealized baroclinic wave simulations described later, utilize E3SM's standard 72 vertical levels (L72).

The E3SMv2 land model (ELMv2) runs on the same grid as the atmospheric physics. ELMv2 upgrades the prescribed vegetation distribution for better consistencies between land use and changes in plant functional types across platforms, and adopts the new shortwave radiation model SNICAR-AD (Dang et al., 2019) for snow and ice. The land use harmonization version 2f data (LUH2; https://luh.umd.edu/data.shtml) (Hurtt et al., 2020) are converted into E3SMv2 plant functional types with an updated version of the land use translator (Di Vittorio et al., 2014). The trajectory of land cover change has also been

improved through better tracking of previous land use change. The E3SMv2 river routing model (Model for Scale Adaptive River Transport, MOSARTv2) utilizes the regular lat-lon grid ($0.5°$ for LR and $0.125°$ for NARRM). MOSARTv2 uses the kinematic wave method to route the runoff from ELM into the ocean model via an eight-direction-based river network (Li et al., 2013). More details about the E3SMv2 model are documented by Golaz et al. (2022).

## 2.1 EAM hybrid timestep strategy for RRM production simulations

In previous RRM studies, including the EAMv1 CONUS RRM (Tang et al., 2019), the atmospheric physics timestep is often chosen to be shorter than that of the globally uniform low-resolution model to match the highest resolution grids in the RRM. Such treatment cannot satisfy the two criteria above for the purpose of global climate production simulations mainly because the ZM deep convection scheme and other cloud parameterizations used by EAM are by design not scale-aware (Xie et al., 2018). If the EAM in NARRM used a shorter physics timestep than LR while keeping other physics parameters unchanged,

the NARRM results on the unrefined portion of the mesh (covering a larger area than the refined portion) would not match the quality of the LR results and thus undermine the NARRM global performance. Furthermore, limited by the poor scale-aware cloud parameterizations, we would not achieve optimal climate over multi-scale RRM grids with a single set of physics parameters even if we opted to retune NARRM. With these considerations, in the present study when employing RRM for climate production campaigns, we opt for a hybrid timestep strategy in EAM, which is a combination of LR physics timestep and the

high-resolution dynamics timesteps (see Table 1). In this way, NARRM retains much of the LR global climate characteristics with possible improvements at the refined area benefiting from the high-resolution dynamics. Moreover, this approach simplifies the RRM development as it naturally avoids further tuning the RRM beyond what was done for LR. This choice also ensures that the physics behaves as closely as possible between the LR and RRM simulations, to facilitate direct comparisons of their climates.





## 2.2 EAM running on unstructured meshes

In EAM, the underlying grid is always treated as fully unstructured. EAM can run on any grid that represents a tiling of the sphere with quadrilateral elements. For quasi-uniform grids, EAM relies on cubed-sphere grids since these grids are simple to construct. RRM grids are constructed by external tools described below. Internally, the code treats all these grids identically, the only difference being the various resolution-dependent parameters. For the dynamical core, these parameters consist of the many timesteps in the model (given in Table 1) and the hyperviscosity coefficient. The dynamical core timesteps are chosen to ensure stability of the model. For RRM grids, these timesteps are chosen to match those that would be used in a global model with the same resolution as the highest resolution contained within the RRM. For the NARRM grid used here, which includes refinement down to 25 km, we use the same timesteps as would be used by a global 25-km configuration of EAM.

**Table 1.** Column numbers and timesteps of the atmosphere component used in LR and NARRM simulations.

| Model | Column # | | Timesteps (s) | | | | Physics |
|-------|----------|---------|---------------|--------|--------------|--------|---------|
| | Dynamics | Physics | Dynamics | | | | |
| | | | Hyperviscosity | Dycore | Dycore Remap | Tracer | |
| LR | 48,602 | 21,600 | 300 | 300 | 600 | 1,800 | 1,800 |
| NARRM | 130,088 | 57,816 | 75 | 75 | 150 | 450 | 1,800 |

For hyperviscosity, EAM relies on a resolution-aware tensor hyperviscosity formulation (Guba et al., 2014) applied on each model surface. The tensor coefficients vary spatially based on the two length scales of each spectral element (derived from the eigenvalues of the reference element map). This operator has a built in scaling of $\Delta x^3$ with strength controlled by a coefficient $\nu$ with units of $s^{-1}$. The tensor is designed to have the proper directional resolution dependence for highly distorted elements, while matching the traditional constant-coefficient hyperviscosity on square elements. In EAMv2, we use the tensor hyperviscosity operator with $\nu = 3.4 \times 10^{-8} s^{-1}$ for all grids (cubed-sphere and RRM) and at all resolutions. The only exception is the LR $1°$ cubed-sphere grid, where for continuity with older simulations we continue to use the constant-coefficient hyperviscosity operator with $\mu = 1 \times 10^{15} m^4/s$. For a uniform degree $p$ spectral element grid with square elements, the tensor operator with coefficient $\nu$ is identical to a constant coefficient hyperviscosity operator with coefficient $\mu = \left(\frac{p}{2}\right)^3 R \Delta x^3 \nu$, where $\Delta x$ is the element edge length divided by $p$ and $R$ is the radius of the sphere. In EAM, we always use $p = 3$.

## 2.3 Key tools for the RRM configuration

A number of tools have been developed to streamline the workflow for EAM/ELM simulations on RRM grids. These are described as follows, in the approximate order they are employed:

- **The Spherical Quadrilateral Grid Generator (SQuadGen):** Generation of the atmosphere/land mesh is performed using SQuadGen (Ullrich, 2022; Guba et al., 2014). This tool translates a monochrome PNG image, which denotes the desired level of grid refinement on an equirectangular projection, to a mesh of refined quadrilaterals based on a cubed-





sphere. The use of quadrilaterals is by necessity for compatibility with the spectral element dynamcial core. Transition regions are managed using "paving", that is, using predefined patterns of quadrilaterals which enable transition between coarse resolution and fine resolution regions. Smoothing of the grid is performed via spring dynamics. The spectral elements of the NARRM grid produced with this procedure are shown in Fig. 1. In the spectral element method, each field is represented by polynomials up to degree 3 within each element. The resolution represented by each element (its average length divided by 3) is shown in Fig. 2.

- **TempestRemap:** The TempestRemap package (Ullrich and Taylor, 2015; Ullrich et al., 2016) is used to generate conservative, consistent, and monotone linear maps between fields stored as volume averages (i.e., updated using the finite volume methods) and fields stored as spectral elements (i.e., as coefficients of a set of basis functions). The generated maps require the construction of an "overlap mesh", which is the union of the source and target face, the generation of an approximate map, and subsequent projection of the approximate map onto the linear space of conservative, consistent, and (optionally) monotone maps.

- **Topography generation:** To generate topography on the NARRM grid, we rely on the tool chain described in Lauritzen et al. (2015) combined with a topography smoothing tool included with HOMME. The latter tool is chosen so that the topography smoothing is done with the same discrete Laplace operator used internally in the dynamical core.

- **NetCDF Operators (NCO):** NCO consists of a number of command-line tools that enable manipulation of netCDF files (Zender, 2008). The tools include variable extraction, remapping, and spatial and temporal averaging. Provenance information is preserved within the netCDF files to enable scientific reproducability.

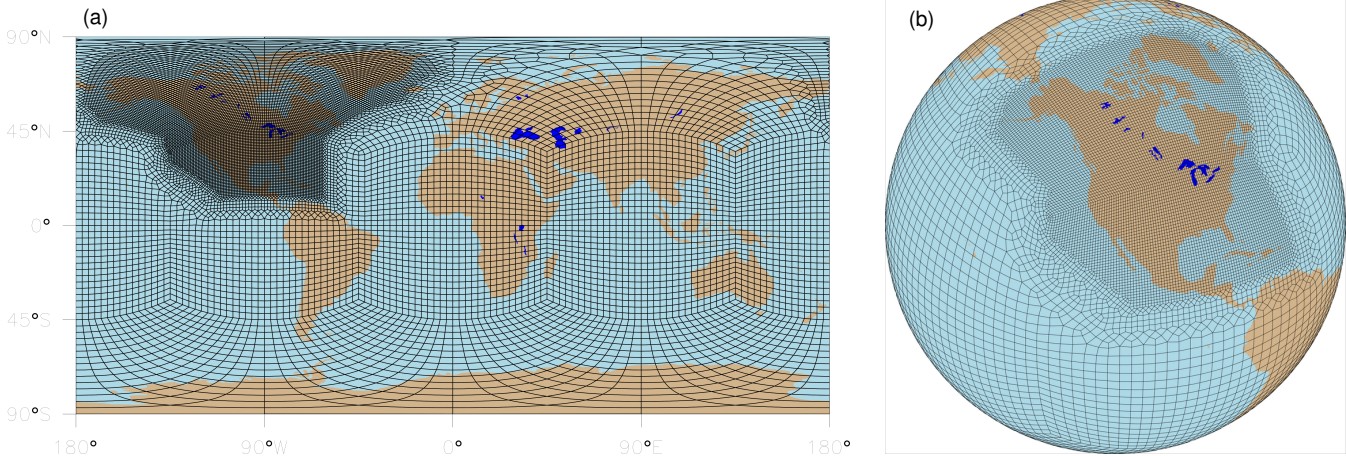

**Figure 1.** North American RRM (NARRM) grids for the atmosphere dynamical core shown in (a) a cylindrical equidistant projection and (b) an orthographic projection.





## 2.4 Idealized test

Before running long coupled NARRM simulations, we first evaluate the dynamical core settings for the NARRM grid using a
baroclinic instability test case. This test case establishes that the dynamical core behaves as expected in an idealized setting: the
timesteps are stable, the model can capture high resolution features in the high resolution region, and the presence of the high
resolution and mesh transition regions do not negatively impact the large scale behavior. For this evaluation, we use an extension
of the dry baroclinic wave test case by Ullrich et al. (2014) with two idealized, analytically-prescribed mountains (Hughes and
Jablonowski, 2022). The latter now serve as the trigger for baroclinic instability. The addition of the two mountains generates a
flow with more energy at smaller scales as compared to the original test case, especially downstream of the mountains, making
this an attractive test case for studying the impacts of resolution.

For this test case, we run simulations with three different horizontal grids, LR, NARRM, and HR, and 30 hybrid vertical
levels (L30) which are specified in Appendix B of Reed and Jablonowski (2012). The LR and NARRM grids are as described
above, and we add an HR grid. The HR grid is a global $0.25°$ grid which matches the high resolution region of the NARRM
grid. All idealized runs use the same settings as in the full model (except L30 instead of L72), with HR and NARRM using
identical timesteps since they both contain regions of $0.25°$ resolution. All simulations utilize the EAMv2 tensor hyperviscosity
tuning with $\nu = 3.4 \times 10^{-8}$ and $\Delta x^3$ resolution scaling.

The test case is fully described in Hughes and Jablonowski (2022). We use the dry configuration and make one modification
to the locations of the mountains. In particular, the center locations of the mountains are shifted longitudinally by $144°$ to the
east in order to place the two mountains within the NARRM's high resolution region. The new center locations are therefore
$144°$W and $76°$W. The peak height of the mountain ranges is 2000 m. Figure 2 illustrates the size and location of the mountains

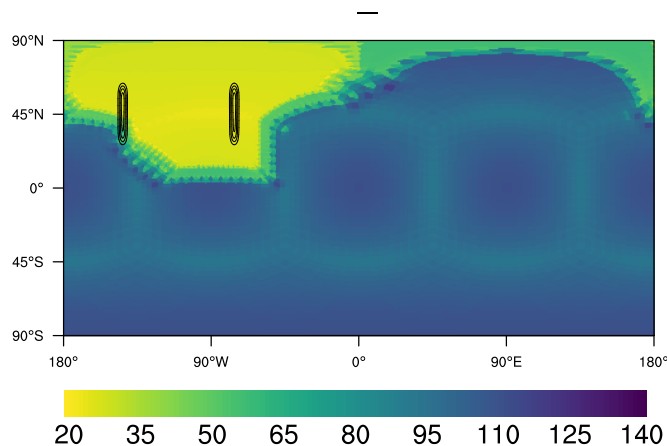

**Figure 2.** Contour lines of the topographic height with a peak amplitude of 2000 m overlaid on a map of the NARRM grid resolution (square
root of element area). The resolution is $\sim$25 km over North America (shown in yellow) transitioning to $\sim$100 km over the rest of the globe
(dark blue). The two mountains are mostly contained within the high-resolution region. In the low resolution region, the faint outline of an
inscribed cube shows the slight nonuniformness of the $1°$ cubed-sphere grid used in that region.



along with the NARRM mesh resolution while Fig. 3 shows the surface pressure at day 6 computed on the same mesh. The latter highlights the topographically generated baroclinic instability in the northern hemisphere.

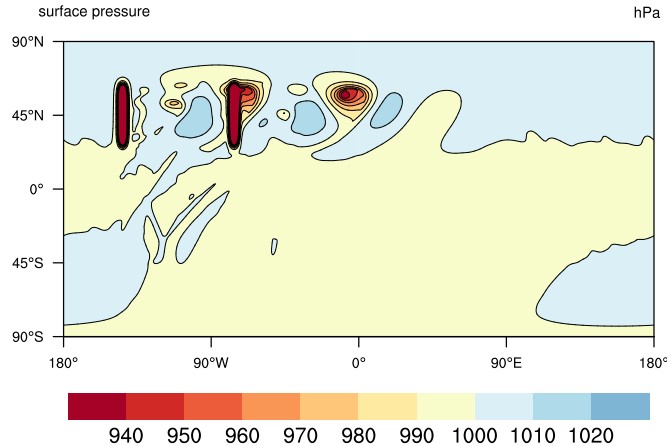

**Figure 3.** Contours of the surface pressure at day 6 showing the topographically triggered baroclinic instability in the northern hemisphere as computed on the NARRM grid. The instability has yet to be triggered in the southern hemisphere. The mountain height contours are overlaid. The colors saturate over the mountain ranges with minimum surface pressure values around 750–780 hPa (not shown).

Figure 4 shows the 750 hPa temperature field after 6 days on all three grids. The plots are zoomed in over the region with the most activity shown in Fig. 3. We first compare the field in the NARRM's high resolution region with the HR result, and note the remarkable agreement between the two solutions in the high resolution region. The presence of high resolution in the NARRM simulation allows the model to capture features in that region with finer scales than can be captured by the LR simulation, as expected. Further downstream from the mountains at the right edge of the figure, the NARRM resolution has transitioned to match the LR resolution and the scales captured by the NARRM solution are no longer as fine as they are in the HR solution. They are somewhat dissipated and fall between the LR and HR results. Thus the presence of the high resolution region in the NARRM grid improves some aspects of the solution in the low resolution region. Finally, we note that there are no visible artifacts from the distorted elements in the mesh transition region. Examination of other fields, such as vorticity (not shown) demonstrate similar results.

## 3   Simulations and computational cost

We perform a set of NARRM production simulations parallel to the LR version documented by Golaz et al. (2022) and following the same CMIP6 specifications. The LR and NARRM production simulations analyzed in the present study are summarized in Table 2. These simulations consist of the CMIP6 Diagnosis, Evaluation, and Characterization of Klima (DECK) and historical simulations (Eyring et al., 2016), i.e., one pre-industrial control (*piControl*, 500 years), two idealized $CO_2$ runs (*1pctCO2* and *abrupt-4xCO2*, each 150 years), a five-member historical ensemble (*historical_N*, 1850–2014), and a



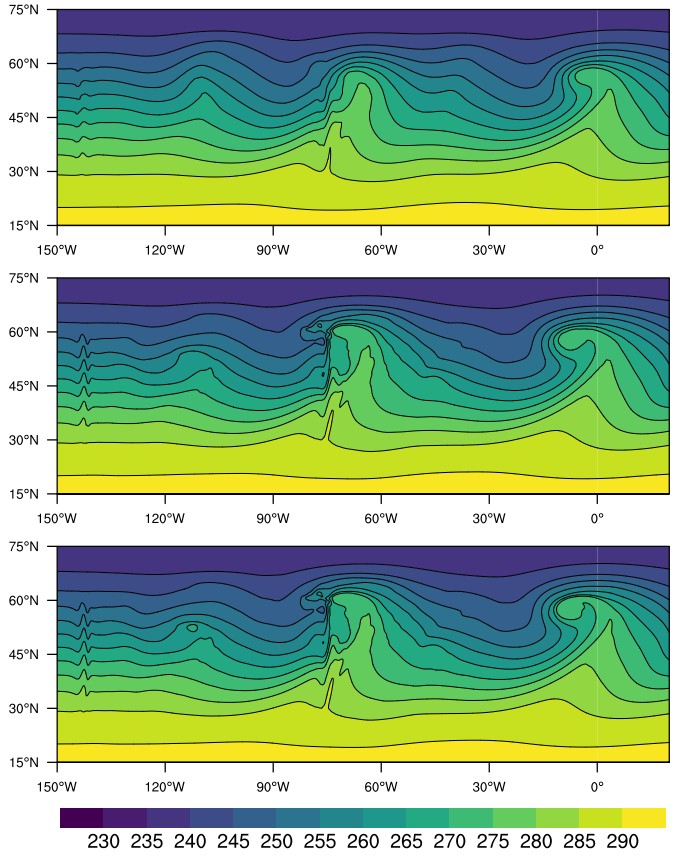

**Figure 4.** Contours of the 750 hPa temperature (degrees K) at day 6, plotted over a subset of the globe containing the mountains and most of the downstream region affected by the baroclinic instability. Results are shown from the LR grid (top), NARRM (middle), and HR grid (bottom).

three-member Atmospheric Model Intercomparison Project (*amip*) type ensemble (*amip_N*, 1870–2014). Initial conditions are taken from January 1 of different years of *piControl* as indicated in Table 2 for *1pctCO2*, *abrupt-4xCO2*, and *historical_N* simulations. *amip_N* simulations are initialized from the 1870 condition of corresponding *historical_N* simulations.

In order to estimate the effective radiative forcing of anthropogenic aerosols in LR and NARRM configurations, we perform pairs of nudged simulations with prescribed emissions of aerosols and their precursors for the present-day (PD, year 2010)

and pre-industrial (PI, year 1850), which are taken from the CMIP6 emission data. Table 3 lists the nudged simulations used to assess the effective radiative forcing of anthropogenic aerosols. Horizontal winds in LR and NARRM are nudged towards wind fields from their respective baseline simulations, with a relaxation time scale of 6h. These nudged simulations are 15-month long, with the first 3 months discarded as spin-up. Previous studies have shown that nudging the horizontal winds can help constrain the large-scale circulation in the model (Zhang et al., 2014; Sun et al., 2019; Tang et al., 2019), so that the

anthropogenic aerosol effects can be determined with relatively short simulations (Zhang et al., 2022b, c).



**Table 2.** Summary of E3SMv2 LR (Golaz et al., 2022) and NARRM production simulations analyzed in this study.

| Label | Description | Period | Ens. | Initialization |
|-------|-------------|--------|------|----------------|
| | **Fully coupled** | | | |
| | (atmosphere, ocean, sea ice, land and river) | | | |
| *piControl* | Pre-industrial control | 500 years | - | Pre-industrial spinup |
| *1pctCO2* | Prescribed 1% yr$^{-1}$ $CO_2$ increase | 150 years | 1 | *piControl* (101) |
| *abrupt-4xCO2* | Abrupt $CO_2$ quadrupling | 150 years | 1 | *piControl* (101) |
| *historical_N* | Historical | 1850-2014 | 5 | *piControl* (101, 151, 201, 251, 301) |
| | **Prescribed SST and sea ice extent** | | | |
| | (atmosphere, thermodynamic sea ice, land and river) | | | |
| *amip_N* | Atmosphere with prescribed SSTs and sea ice concentration | 1870-2014 | 3 | *historical_N* (1870) |

**Table 3.** Nudged LR and NARRM atmospheric model simulations used in this study. All simulations are performed with prescribed sea surface temperature (SST) and sea ice concentration for year 2010. Nudging data are 6-hourly model output saved from the LR and NARRM free-running simulations (middle column). Due to the model instability problem with nudging application in RRM (with a relatively long time step), we use an alternative physics-dynamics coupling approach (see option "se_ftype = 1" in section 3.1 of Zhang et al. (2018)) for the NARRM nudged simulations. We find the impact of using different physics-dynamics coupling approaches on the global mean effective aerosol forcing estimate in LR to be small (difference < 0.05 W m$^{-2}$).

| Label | Baseline simulation | Emission |
|-------|---------------------|----------|
| *Nudge_LR_PD* | LR | 2010 |
| *Nudge_LR_PI* | LR | 1850 |
| *Nudge_NARRM_PD* | NARRM | 2010 |
| *Nudge_NARRM_PI* | NARRM | 1850 |





### 3.1 Computational performance

A sequence of performance benchmark simulations were run on the Argonne National Laboratory Chrysalis cluster. Chrysalis has 512 compute nodes. Each node has two AMD Epyc 7532 "Rome" 2.4 GHz processors. Each processor has 32 cores, for a total of 64 cores per node. Each node has 256GB 16-channel DDR4 3200MHz memory. The interconnect hardware is

Mellanox HDR200 InfiniBand and uses the fat tree topology. The model code was compiled using Intel release 20200925 with GCC version 8.5.0 compatibility and run using OpenMPI 4.1.3 provided in the Mellanox HPC-X Software Toolkit.

The simulations are run with one MPI process per core and no OpenMP threading. Throughput values are computed using the maximum wall-clock time (minimum throughput) over all MPI processes; model initialization time is excluded. A throughput data point corresponds to one simulation run for 90 days. The input/output (I/O) configuration is identical to production

simulations. At the end of 90 days, a restart file is written.

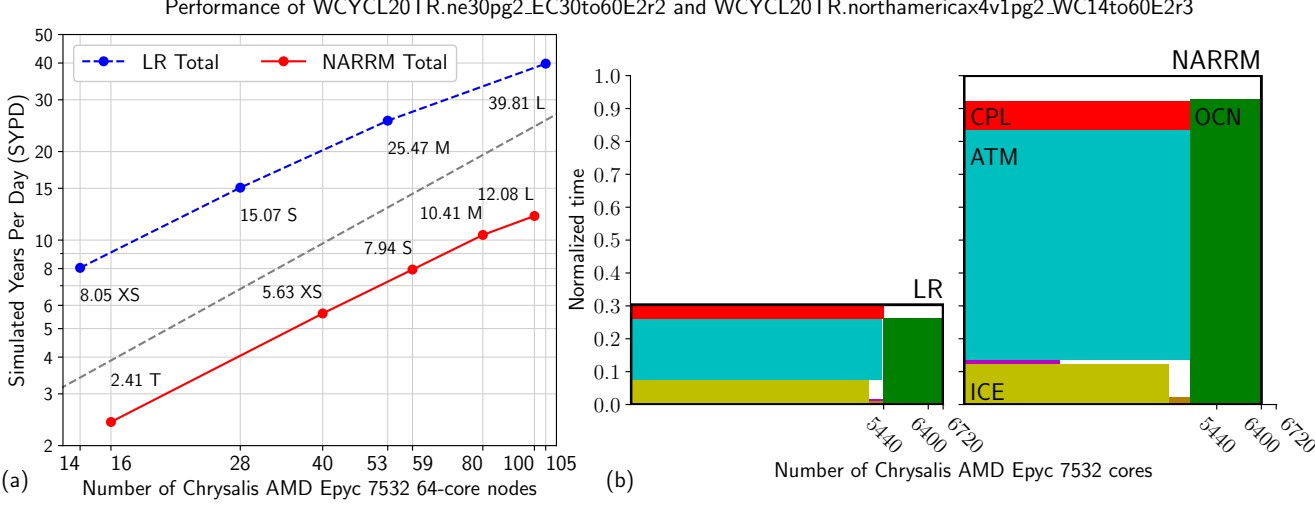

**Figure 5.** Performance of the LR and NARRM historical simulations. (a) Throughput vs. number of computer nodes. Each data point is annotated with its throughput in simulated years per day (SYPD) and computer resource configuration name. The dashed gray line shows the perfect-scaling slope. (b) Computational resource plots for the L process layouts. Each component has one rectangle. A rectangle has the area given by the product of normalized wall-clock time and number of cores, with the NARRM total time normalized to 1.0.

Figure 5 summarizes the performance of the LR and NARRM *historical_N* simulations for several node counts and corresponding process layouts with names T (NARRM only), XS, S, M, and L. Note that while the layout names are shared among models, the specific layout associated with a name differs among models. Each simulation's data point is annotated with its throughput in simulated years per day (SYPD) and process layout name. The highest throughput of the LR simulations is 39.81

SYPD. In Golaz et al. (2022, Fig. 2), the highest throughput is 41.89 for the same node count; *historical_N* simulations have additional forcings to compute relative to the *piControl* simulation used in Golaz et al. (2022, Fig. 2). The LR throughput falls



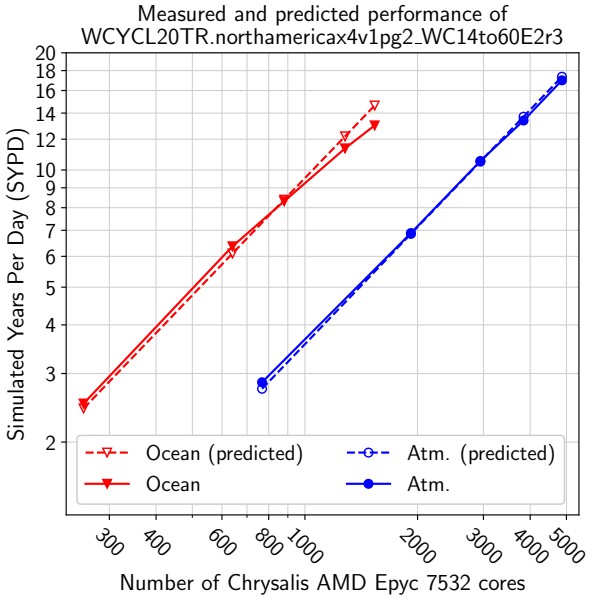

**Figure 6.** Performance of the atmosphere (Atm.) and ocean components of the NARRM historical simulation. Solid lines show measured performance. Dashed lines show the performance predicted by a simple model that uses the LR simulation with the XS process layout for input data; see the text for a description of the performance model.

off from the perfect scaling slope faster than the NARRM throughput because the LR simulation has less work per node. For the L process layouts, and accounting for 105 vs. 100 nodes, the throughput factor difference is 3.14.

Figure 5b shows the wall-clock-time–resource product for each component for the L layouts. A rectangle's width is proportional to the number of cores the component uses; its height is proportional to the wall-clock time to simulate a fixed simulation period, with the time normalized so that the NARRM simulation has a total time of 1.0. The atmosphere (ATM), sea ice (ICE), coupler (CPL), land (LND), and river runoff (ROF; LND and ROF are too small to label) components run on one set of nodes, while the ocean (OCN) component runs on another set. An unfilled rectangle having "LR" or "NARRM" at the top-right corner shows the total product. Because there is no global communication barrier between components run in sequence, the time value

of each component is approximate, and, thus, the filled rectangles do not sum to the total time.

        We can understand the NARRM component-level performance as a function of spatial and temporal discretization parameters and one LR simulation to calibrate throughput. The LR calibration simulation should reflect that RRM simulations have a large amount of work per node; thus, we use the LR simulation run with the XS process layout, the left-most LR point in Fig. 5(a). We focus on the two most expensive components, the atmosphere and ocean. We start with the ocean, whose performance is

simpler to model. For simplicity, we write the formulas in terms of wall-clock time (w.c.t.) for a fixed simulation length, e.g., 90 days. The input measured datum is the top-level ocean-component (ocn) wall-clock time in the LR simulation run with the XS process layout, w.c.t.$_{\mathrm{LR}}^{\mathrm{ocn}}$. The input parameters are the number of computer cores ($n_{\mathrm{core}}$) used in the LR (XS) and RRM



(variable) simulations, the number of cells ($n_{\text{cell}}$) in each grid, and the time steps ($\Delta t$) in each simulation. For a fixed simulation length, the predicted ocean-component RRM performance is then

$$\text{w.c.t.}_{\text{RRM}}^{\text{ocn}} = \frac{(n_{\text{core}})_{\text{LR}}^{\text{ocn}}}{(n_{\text{core}})_{\text{RRM}}^{\text{ocn}}} \cdot \frac{(n_{\text{cell}})_{\text{RRM}}^{\text{ocn}}}{(n_{\text{cell}})_{\text{LR}}^{\text{ocn}}} \cdot \frac{(\Delta t)_{\text{LR}}^{\text{ocn}}}{(\Delta t)_{\text{RRM}}^{\text{ocn}}} \cdot \text{w.c.t.}_{\text{LR}}^{\text{ocn}}. \tag{1}$$

The performance model for the atmosphere is more complicated because it has two important time steps, one each for the dynamical core (dynamics) and the column parameterizations (physics). Thus, the factor accounting for model time steps is broken into two terms, one each for the physics and dynamics. The predicted atmosphere-component RRM performance is then

$$\text{w.c.t.}_{\text{RRM}}^{\text{atm}} = \frac{(n_{\text{core}})_{\text{LR}}^{\text{atm}}}{(n_{\text{core}})_{\text{RRM}}^{\text{atm}}} \cdot \frac{(n_{\text{cell}})_{\text{RRM}}^{\text{atm}}}{(n_{\text{cell}})_{\text{LR}}^{\text{atm}}} \cdot \left( \frac{(\Delta t_{\text{physics}})_{\text{LR}}^{\text{atm}}}{(\Delta t_{\text{physics}})_{\text{RRM}}^{\text{atm}}} \cdot \text{w.c.t.}_{\text{LR}}^{\text{atm physics}} + \frac{(\Delta t_{\text{dynamics}})_{\text{LR}}^{\text{atm}}}{(\Delta t_{\text{dynamics}})_{\text{RRM}}^{\text{atm}}} \cdot \text{w.c.t.}_{\text{LR}}^{\text{atm dynamics}} \right). \tag{2}$$

Figure 6 shows the results of these models, where wall-clock time and simulation length have been converted to throughput (SYPD). The solid lines show the measured throughput of each component as a function of number of computer cores. The dashed lines show the corresponding throughput values predicted by Eqs. 1 and 2 with, again, the LR XS ocean and atmosphere throughputs as the input measured data. The primary error in the model is not accounting for a fall-off in scaling at large core counts. Because this fall-off is small for the atmosphere and ocean components, these simple performance models are accurate and can be used to predict the cost of other model configurations. For example, a uniform high-resolution atmosphere model would use the ne120pg2 grid. Using the same time steps and number of vertical levels as in the NARRM configuration, for fixed computational resources, the high-resolution atmosphere configuration's throughput would be $6 \cdot 120^2/14454 = 5.98$ times smaller than the NARRM configuration's throughput, where this factor is the quotient of the numbers of elements in each of the two grids.

## 4   Global climate

As described above, the RRM model is expected to simulate a global climate similar to the LR model for production simulation campaigns since most areas are still covered by the same LR grids. In this section, we will examine whether this is the case for the global mean climate, climate sensitivity, as well as climate feedback.

For the global climatology, we focus on the last three decades (years 1985–2014) of historical simulations when more observational datasets are available. Figure 7 provides an overall comparison of the global mean climate among LR (blue triangles), NARRM (red triangles), and CMIP6 (boxes and whiskers) models as quantified by the uncentered spatial root-mean-square error (RSME) relative to the observations or reanalysis data. The RSME numbers are calculated with the E3SM Diags package (Zhang et al., 2022a) for the first historical member (0101 for LR and NARRM, and r1i1p1f1 for CMIP6 models). Figure 7 clearly shows that NARRM and LR simulate very similar annual and seasonal averages. NARRM outperforms LR in the June-July-August (JJA) shortwave (SW) cloud radiative effect (CRE) partly because it better represents low clouds in NA (see Fig. 12 for the example at California). NARRM also simulates slightly better December-January-February (DJF) precipitation compared to LR, partly due to its improved topography (Fig. A1) and orographic precipitation in NA (see Fig. 11bdf). On



**Figure 7.** Comparison of the global spatial RMSE of model climatology (annual and seasonal averages of years 1985–2014) vs. observations with the E3SM Diags package (Zhang et al., 2022a). The model results are from the first historical member of E3SMv2 (0101) LR (blue triangles) and NARRM (red triangles), and 52 CMIP6 models (r1i1p1f1). The boxes and whiskers show 25th, 75th percentile, minimum and maximum RMSE of the CMIP6 ensemble. Quantities include (a) TOA net radiation flux, (b, c) TOA SW and LW cloud radiative effects, (d) precipitation, (e) surface air temperature over land, (f) sea-level pressure, (g, h) 200- and 850-hPa zonal wind, and (i) 500-hPa geopotential height. TOA = top-of-atmosphere; SW = shortwave; CRE = cloud radiative effects; LW = longwave; ANN = annual; DJF = December–February; MAM = March–April; JJA = June–August; SON = September–November; RMSE = root-mean-square error. The climatology of the observations and reanalysis data are calculated from CERES-EBAF Ed4.1 (Loeb et al., 2018) (2001–2014) for (a, b, and c), GPCP2.3 (Adler et al., 2018) (1985–2014) for (d) and ERA5 (Hersbach et al., 2020) (1985–2014) for (e, f, g, and h).



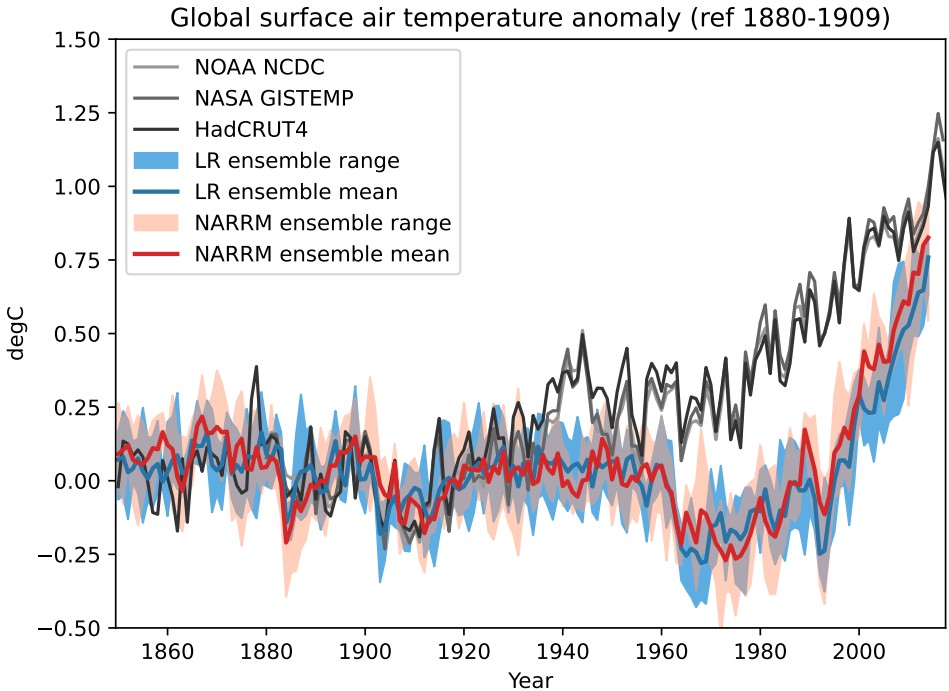

**Figure 8.** Time series of global annual mean surface air temperature anomalies from the ensemble mean of LR (blue) and NARRM (red) historical runs and observational datasets (grays) (National Oceanic and Atmospheric Administration (NOAA) National Climatic Data Center (NCDC), National Aeronautics and Space Administration (NASA) GISTEMP, and HadCRUT4). The model ensemble minimum-maximum ranges are shaded.

the other hand, NARRM does not perform as well as LR for some other fields, such as the 200-hPa zonal wind in JJA and
September-October-November (SON), which are associated with the increased positive biases in the tropical western Pacific and Amazon (not shown).

Figure 8 compares the long time series (years 1850–2014) of global annual average anomalies in the surface air temperature from the ensemble means of LR and NARRM historical simulations, and observational datasets (National Oceanic and Atmospheric Administration National Climatic Data Center (Smith et al., 2008; Zhang et al., 2015), National Aeronautics and
Space Administration GISTEMP (GISTEMP Team, 2018; Hansen et al., 2010), and HadCRUT4 (Morice et al., 2012)). Over the whole period, NARRM tracks LR closely, including good agreement with observations until the 1930s and low biases afterwards, which are mainly attributed to too strong aerosol-related forcing and feedback (see Golaz et al., 2022, for details). This is further confirmed by the fact that the global mean effective radiative forcing of anthropogenic aerosols in NARRM and LR are very similar (-1.415 W/m$^2$ versus -1.421 W/m$^2$) as quantified by a pair of nudged simulations (see Section 5.1.2).



**Figure 9.** Comparison of climate sensitivities between LR (left) and NARRM (right) derived from idealized $CO_2$ forcing simulations. (a, b) time series of global annual mean surface air temperature anomaly from the following simulations, *abrupt-4xCO2* (red), *1pctCO2* (blue), and the control (*piControl*; green). The transient climate response (TCR) is computed as a 20-year average around time of $CO_2$ doubling (year 70). (c, d) Gregory regression plots. The estimated effective climate sensitivity (ECS) and effective 2x $CO_2$ radiative forcing (F) are as labeled.





Following the CMIP6 DECK protocol (Eyring et al., 2016), we quantify the climate sensitivity and feedback with the abrupt quadrupling of $CO_2$ (*abrupt-4xCO2*) and the transient climate response (TCR) with a simulation forced by a 1% $yr^{-1}$ $CO_2$ increase (*1pctCO2*) relative to the pre-industrial control simulation (*piControl*). The equilibrium climate sensitivity (ECS) is estimated with the linear regression of top-of-atmosphere (TOA) radiation change against surface temperature change in a 150-year *abrupt-4xCO2* simulation (Gregory et al., 2004). The 2x$CO_2$ effective radiative forcing (ERF) is computed as the Y-

intercept of the Gregory plot divided by two, which measures the energy imbalance caused by doubling the atmospheric $CO_2$ concentration while keeping the surface temperature unchanged. TCR, which measures the response on shorter time scales, is derived based on its definition — the average surface temperature change in the 20-year period when the $CO_2$ concentration doubles from a *1pctCO2* experiment.

    Figure 9 depicts the annual mean surface temperature change as a function of time and the Gregory plots from the idealized

$CO_2$ experiments with LR and NARRM. The differences in climate sensitivity between LR and NARRM are very subtle as quantified by both ECS (4.00 K vs. 3.94 K) and TCR (2.41 K vs. 2.44 K).

    The regression slope in the Gregory plot (Fig. 9cd) denotes the total radiative feedback caused by the quadrupled $CO_2$ concentration. We further apply the radiative kernel method (Soden et al., 2008; Held and Shell, 2012) to decompose the total radiative feedback into non-cloud and cloud feedbacks. The cloud feedback is estimated by adjusting the cloud radiative

effect anomalies for non-cloud influences. Overall, NARRM shows a slightly larger ERF (3.22 W/m$^2$ vs. 2.98 W/m$^2$), which accompanied with the similar ECS produces a stronger negative total climate feedback in NARRM. The total climate feedback is $-0.74$ W/m$^2$/K for LR and $-0.82$ W/m$^2$/K for NARRM, which mainly relates to the slightly weaker positive SW cloud feedback in NARRM than in LR (see Fig. 23).

    In summary, the results in this section confirm that NARRM with the hybrid timestep methodology simulates largely identical

global climate as its corresponding LR configuration and hence satisfies one of the necessary requirements of global RRM production simulations we proposed in the introduction.

## 5   North American results

In this section, we will zoom in over the refined region over North America (NA) and emphasize climate aspects most relevant to the E3SM water cycle scientific goals (Leung et al., 2020) as well as some weaknesses in LR revealed by Golaz et al. (2022).

The results will be described for the atmosphere, land, and river models, respectively. Moreover, we will analyze interactions between different components (i.e., land-atmosphere coupling), because these interactions are also expected to change with the resolution increase.





## 5.1 Atmosphere

### 5.1.1 Hydrology over the CONUS

First, we look at the overall atmospheric results over the CONUS ($20°– 50°$N, $65°– 125°$W) by comparing the spatial RMSEs of the historical ensemble means between LR (blue triangles) and NARRM (red triangles) in Fig. 10. The same metric is used in Fig. 7 for the global results, but we adjust the variables to be more relevant to the CONUS. NARRM generally produces better (as quantified by smaller RMSE numbers) results than LR for these annual and seasonal climatologies, such as SW CRE (Fig. 10a), precipitation (Fig. 10c), and 200-hPa zonal wind (Fig. 10f). Because we have not retuned the physics of NARRM,

some deteriorations are expected, for example longwave (LW) CRE in DJF and March-April-May (MAM) (Fig. 10b).

Precipitation and clouds, which are obviously important quantities for the water cycle, are largely improved in NARRM compared to LR. The precipitation patterns are better captured by NARRM as observed at the Sierra Madre Occidental in JJA (Fig. 11 left column) and in the western US in DJF (Fig. 11 right column) due to better resolved topography in NARRM. The poor representation of marine stratocumulus clouds is a long-standing problem that plagues many ESMs (e.g. Bogenschutz

et al., 2022). The underestimation of summertime low clouds (manifested as the excessive TOA short-wave CRE) in the California stratocumulus region are substantially improved with NARRM (Fig. 12). This stratocumulus improvement is probably because the coupled NARRM has finer ocean grid cells along the NA coastal area, which helps reduce the warm biases in the sea surface temperature (see Van Roekel et al., 2022, for details), and highlights the advantage of coupled RRM over a single-component RRM.

Another well-known issue of global ESMs is the poorly captured diurnal propagation of organized mesoscale convective systems (MCSs). Over CONUS, such MCSs originate from the front range of the Rockies in the afternoon and propagate eastward, manifesting as a nocturnal precipitation peak in the central US and contributing as much as half of the summertime rainfall in that region (Riley et al., 1987; Jiang et al., 2006). Both LR and NARRM simulate the summertime nocturnal rain peak in the central US (Fig. 13) because of the new convective trigger method for deep convection (Xie et al., 2019). However,

the magnitude is weaker and the area of nocturnal peak extends much larger (almost the whole east half of the US) than the observation, which could be caused by remaining propagation or convective trigger deficiencies. Nevertheless, NARRM reduces the underestimation of maximum diurnal cycle magnitude by over 2x (6.71 mm/day vs. 3.10 mm/day). This result suggests that a resolution of ∼25 km is not adequate to capture the physics driving propagating MCSs, which probably require convection-permitting atmospheric simulations to achieve a good agreement with observations (Caldwell et al., 2021).

### 5.1.2 Aerosols


The E3SMv2 LR model (Golaz et al., 2022) simulates too strong aerosol-related forcing, which has been identified as the primary cause of the underestimated warming in the later portion of historical period in Fig. 8. We will examine here if the NARRM configuration helps bring down the biases in aerosols and anthropogenic forcing by better resolving the meteorological and climate fields. In addition, since NARRM employs the hybrid timestep approach that eliminates re-tuning the

scale-dependent aerosol parameters used in LR, i.e., emission factors of the climate state-dependent or natural aerosols (dust





**Figure 10.** Same as Fig. 7, but contrasting LR and NARRM historical ensemble means at the refined CONUS area (20°– 50°N, 65°– 125°W). Note that variables shown are adjusted to be more appropriate for the CONUS. (a, b) TOA SW and LW cloud radiative effects, (c) precipitation, (d) total precipitable water, (e) surface air temperature, (f, g) 200- and 850-hPa zonal wind, and (h, i) 500-hPa geopotential height and vertical velocity (pressure). The same reference climatology data are used for the variables also shown in Fig. 7, whereas ERA5 (Hersbach et al., 2020) (1985-2014) for (d, e, and i).



**Figure 11.** Comparison of CONUS JJA (left) and DJF (right) precipitation geographic patterns from ERA5 reanalysis (a, b), LR (c, d), and NARRM (e, f) historical ensemble means

and sea salt), we will also discuss the impact of increasing model horizontal resolution on the natural aerosols and total aerosol optical depth (AOD) in NARRM.

1. Impact on anthropogenic aerosols

Aerosols in the NARRM configuration are represented in the same manner as in the LR with the enhanced MAM4 (Wang et al., 2020) and improved dust aerosol properties (Feng et al., 2022). Anthropogenic and wildfire emissions used in the LR and NARRM experiments are also from the same input datasets. However, cloud microphysical processes, horizontal advection, and convection can affect aerosol loading within the NARRM high-resolution domain if wet deposition and/or transport are substantially different from the LR configuration (Caldwell et al., 2019). We first compare modeled surface mass concentrations of $SO_2$, sulfate, black carbon, and organic carbon between the ensemble means of LR and NARRM historical simulations, and evaluate them against ground-based observations from the Clean Air Status



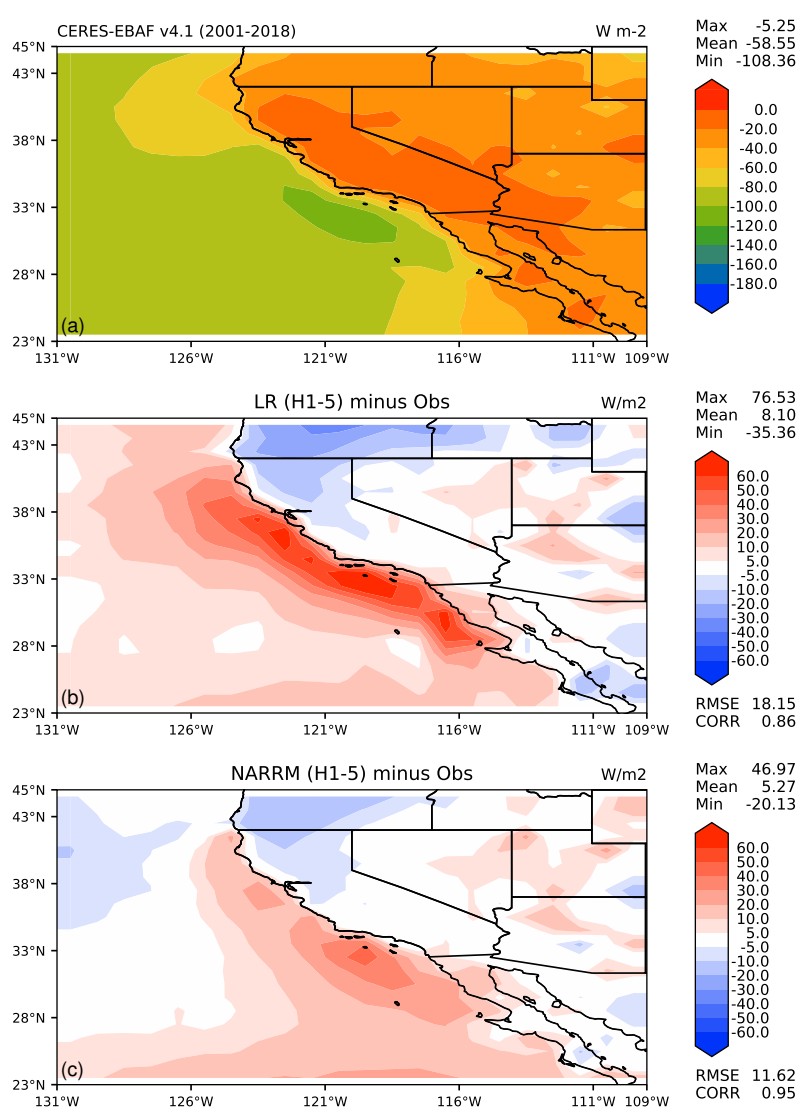

**Figure 12.** Mean TOA short-wave cloud radiative effects at California in JJA of (a) observations (CERES-EBAF Ed4.1), (b) LR (H1-5) minus observation, and (c) NARRM (H1-5) minus observation



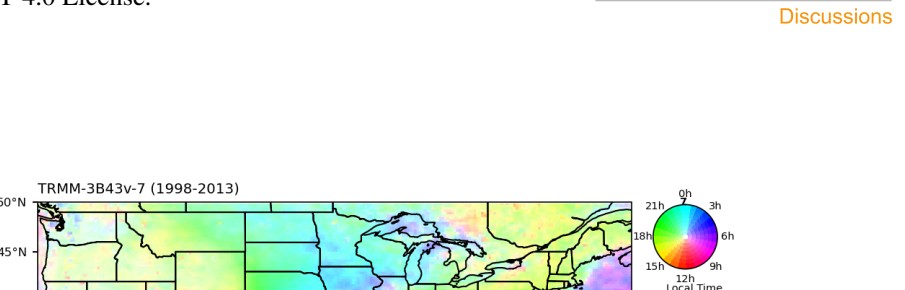

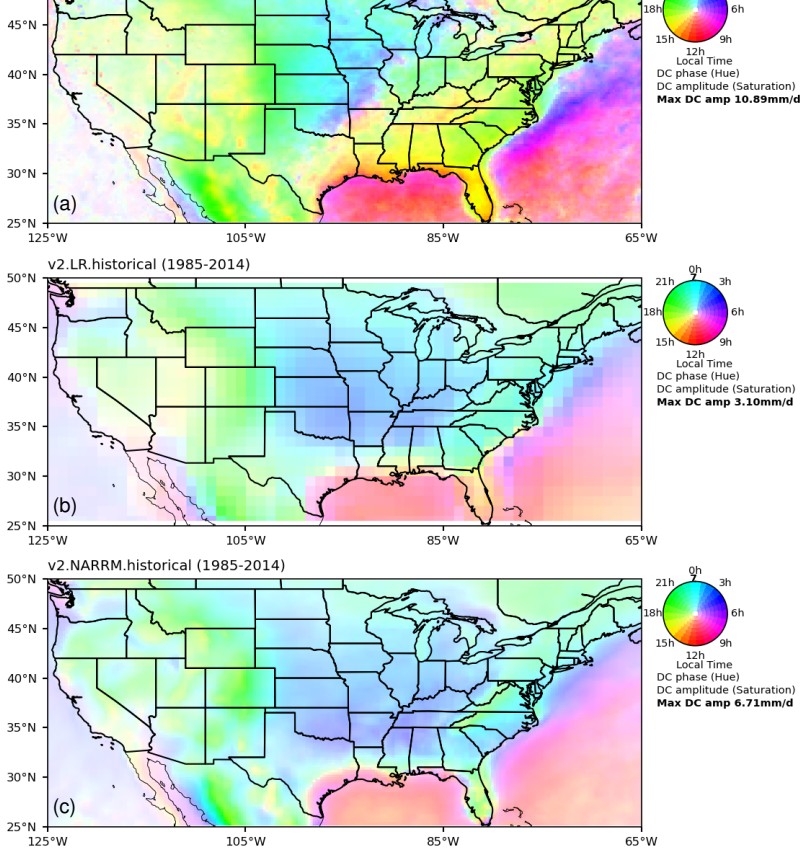

**Figure 13.** Mean diurnal phase (local time, colors) and magnitude (color density) of the maximum precipitation in JJA calculated from the first harmonic of 3-hourly total precipitation (mm/day) for (a) Tropical Rainfall Measuring Mission (TRMM) observations (Huffman et al., 2007), (b) LR (H1-5), and (c) NARRM (H1-5).

and Trends Network (CASTNET) and the Interagency Monitoring of Protected Visual Environments (IMPROVE). The results are shown in Fig. 14. In general, both sets of simulations show a strong correlation with measurements and biases are very similar between the two for the anthropogenic aerosol species. NARRM simulations slightly increase (less than 7%) the concentrations of the four species shown here, compared to LR simulations, which may result from less wet removal or vertical transport in the refined mesh.

2. Impact on natural aerosols

In addition to aerosol removal, emissions of natural aerosols such as dust and sea salt are highly sensitive to resolution changes due to their strong dependence on the resolved surface wind speeds in the model. Increasing model horizontal resolution normally requires re-tuning the dust and sea salt aerosol emission factors. For E3SMv1, Feng et al. (2022)



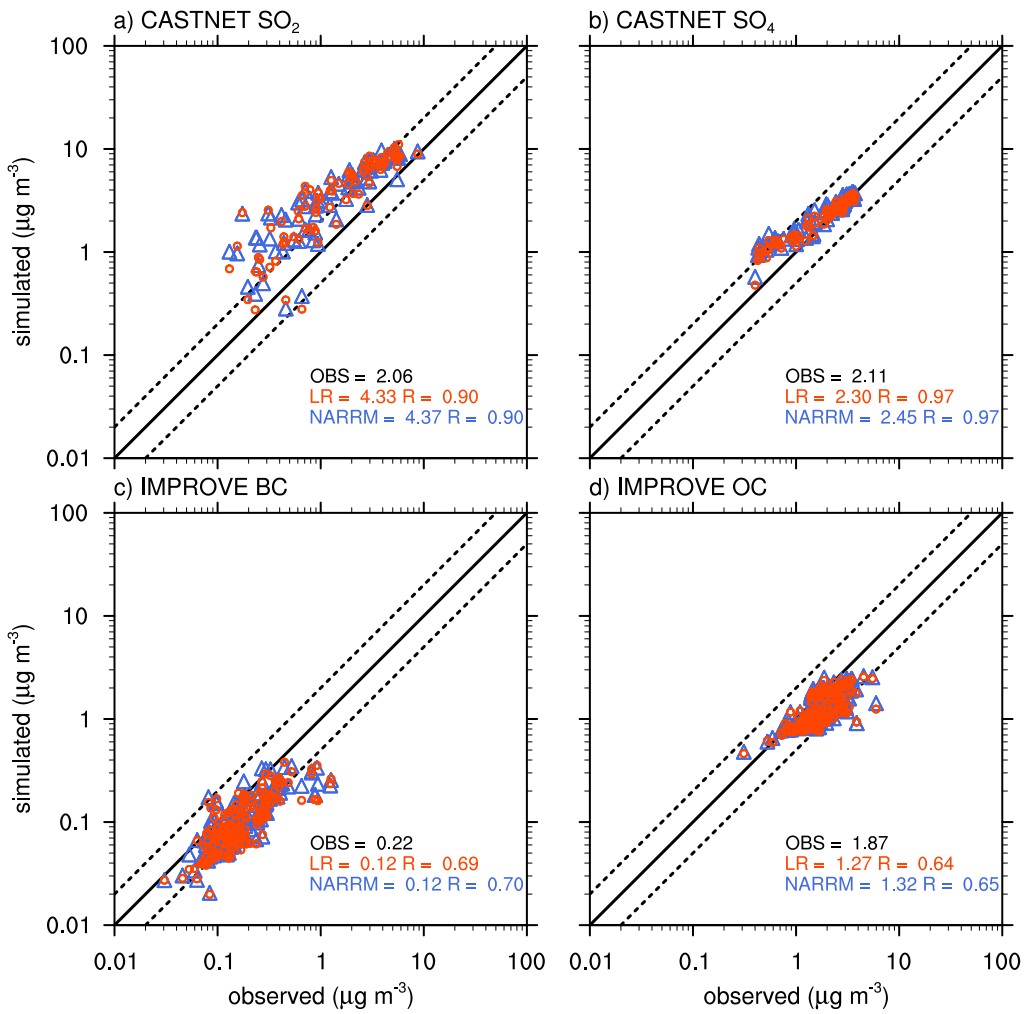

**Figure 14.** Scatter plots of modeled annual mean surface concentrations of (a) SO₂, (b) sulfate, (c) black carbon, and (d) organic carbon (POM+SOA) from LR (H1-5) and NARRM (H1-5) compared to observations at CASTNET and IMPROVE network surface sites during 2005-2014. The numbers are mean concentration and correlation coefficient (R) for data at the individual sites.



**Table 4.** Comparison of simulated annual mean AOD (550 nm) between LR (H1-5) and NARRM (H1-5) for the time period of 1985–2014.

| AOD | Total | Dust | Sea salt | Sulfate | POM[a] | BC[a] | SOA[a] |
|---|---|---|---|---|---|---|---|
| Global means | | | | | | | |
| LR | 0.164 | 0.028 | 0.049 | 0.033 | 0.009 | 0.006 | 0.039 |
| NARRM | 0.163 | 0.028 | 0.049 | 0.033 | 0.009 | 0.006 | 0.038 |
| CONUS means | | | | | | | |
| LR | 0.129 | 0.0098 | 0.0205 | 0.0507 | 0.0074 | 0.0057 | 0.034 |
| NARRM | 0.129 | 0.0101 | 0.0214 | 0.0502 | 0.0075 | 0.0057 | 0.034 |

[a] POM (particulate organic matter), BC (black carbon), and SOA (secondary organic aerosol)

showed that without re-tuning, an increase of the horizontal resolution by a factor of four (i.e., from ∼100 km in LR to ∼25 km in HR) results in about 29% increase of global dust emissions and an even larger increase of dust AOD by 42% due to the combined effects from the weakened removal. In contrast, as shown in Table 4, NARRM historical runs simulate nearly the same global mean AODs as LR for all the aerosol species including dust and sea salt, without changing their emission factors. Over the regionally refined CONUS, the mean dust and sea salt AODs are slightly

increased (<5%). This suggests that NARRM largely retains the performance of LR for the aerosol simulations on the global and regional mean basis, without requiring additional re-tuning of the scale-dependent emission factors.

3. Aerosol spatial variability and extremes

On the other hand, NARRM shows improvement over LR in representing aerosol spatial variability and extreme values over the refined mesh region. Figure 15 compares the simulated AOD (550 nm) distributions between LR (0101) and

NARRM (0101) historical simulations for the present-day time period of 2000–2014. While both depict a similar general geographical pattern, e.g., higher AODs over the more polluted eastern US than the western part of the country, NARRM captures greater and finer detail in spatial variability than LR, e.g., over the mountainous areas along the Rockies, Sierra Nevada, and Appalachians. The better resolved AOD variability in NARRM results from the spatial refinement of the resolution-dependent aerosol emission fluxes (natural species), transport, and removal as discussed above. Compared to

the ground-based AOD measurements at the 37 AERONET (Holben et al., 1998) sites (2006–2015), NARRM shows stronger spatial correlation with the observations than LR (Fig. 15c). Both configurations overestimate the mean AOD averaged over the AERONET sites, possibly linked to the weak wet removal in E3SMv2 (Golaz et al., 2022).

In addition to the improved spatial variability, higher resolution in NARRM also leads to more frequent occurrences of large AOD predictions over CONUS than the LR model, especially over the regions dominated by wind-driven dust

or sea salt aerosols. Figure 16 shows an example of the calculated probability density function (PDF) for dust AOD over the major dust source region in the US (32–42°N, 118–108°W; indicated by the 10°x10° box in Fig. 15b), from both the LR (0101) and NARRM (0101) simulations in 2000–2014. Clearly, NARRM predicts more occurrences of high dust AOD over this region than LR, e.g., 22% of the dust AODs predicted by NARRM exceeds 0.015, which is the



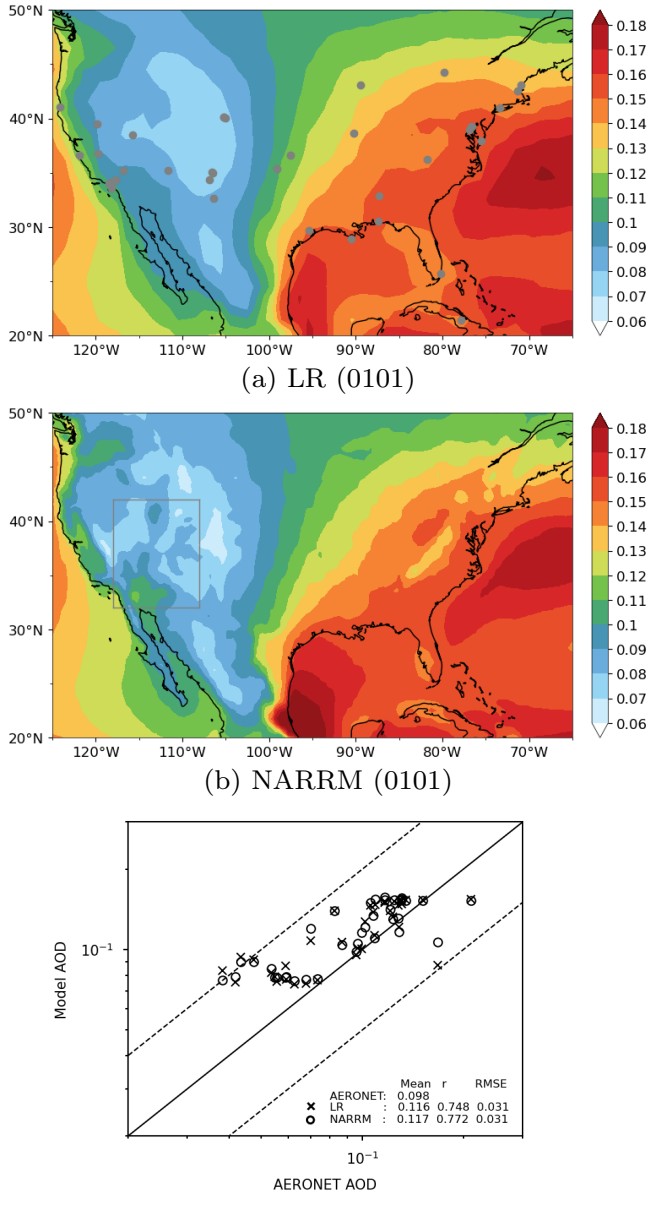

(a) LR (0101)

(b) NARRM (0101)

(c) Comparison with the AERONET AOD

**Figure 15.** Aerosol optical depth (AOD) at 550 nm from (a) LR (0101) and (b) NARRM (0101) historical simulations averaged over 2000–2014. Panel (c) shows the AOD comparison of the two model simulations with the AERONET observations during 2006–2015. The site locations of AERONET are denoted by the gray dots in panel (a). The gray box in panel (b) denotes the dust region referenced in Fig. 16.





top 98th percentile of the LR model predictions. This suggests that LR may significantly underestimate the occurrences

of large dust outbreaks in the southwestern US region relative to NARRM, due to the unresolved surface winds for

dust mobilization in the model. Similarly, NARRM would be more suitable for urban climate or air quality studies for

capturing the extreme polluted cases occurring at finer spatial or temporal scales.

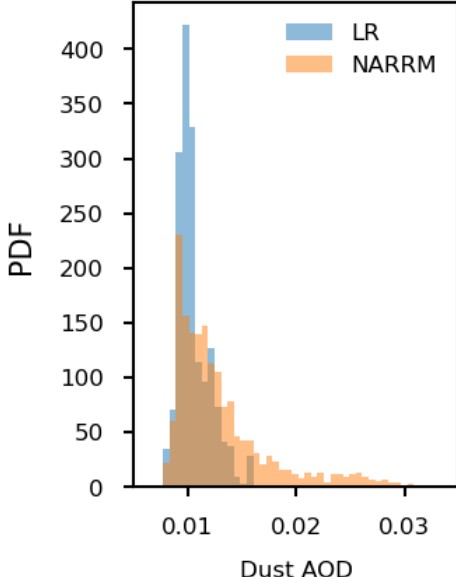

**Figure 16.** Calculated probability density function (PDF) of the dust AOD predictions from LR (0101) and NARRM (0101) between 2000–2014, over the major dust source region in US [32–42°N, 118–108°W; indicated by the 10°x10° box in Fig. 15(b)].

4. Effective radiative forcing of anthropogenic aerosols

Figure 17 shows the effective radiative forcing of anthropogenic aerosols ($\Delta F$) over CONUS and adjacent ocean areas

estimated using nudged LR and NARRM simulations. $\Delta F$ is overall negative in both LR and NARRM and dominated

by the shortwave component ($\Delta F_{SW}$ in Figure 17b,e). The regional mean $\Delta F_{SW}$ and $\Delta F_{LW}$ are both slightly stronger

(more negative for $\Delta F_{SW}$ and more positive for $\Delta F_{LW}$) in NARRM compared to LR. Over the Pacific ocean near

20°N and 120°W, $\Delta F_{SW}$ and $\Delta F$ in NARRM are much stronger than in LR. This is mainly caused by larger low cloud

fraction simulated in NARRM (see Appendix Fig. A3), which causes a larger contrast in droplet number concentration

and liquid water path between the PD and PI simulations compared to LR (Fig. A4).

### 5.1.3   North polar region

The Arctic region has experienced more substantial climate changes in the past decades than the rest of the world (Cohen et al., 2020). Besides mean climate, it is at least equally important for the CMIP6 production simulations to capture such large regional climate variabilities with high fidelity. Clouds play an important role in modulating the surface radiative energy budget





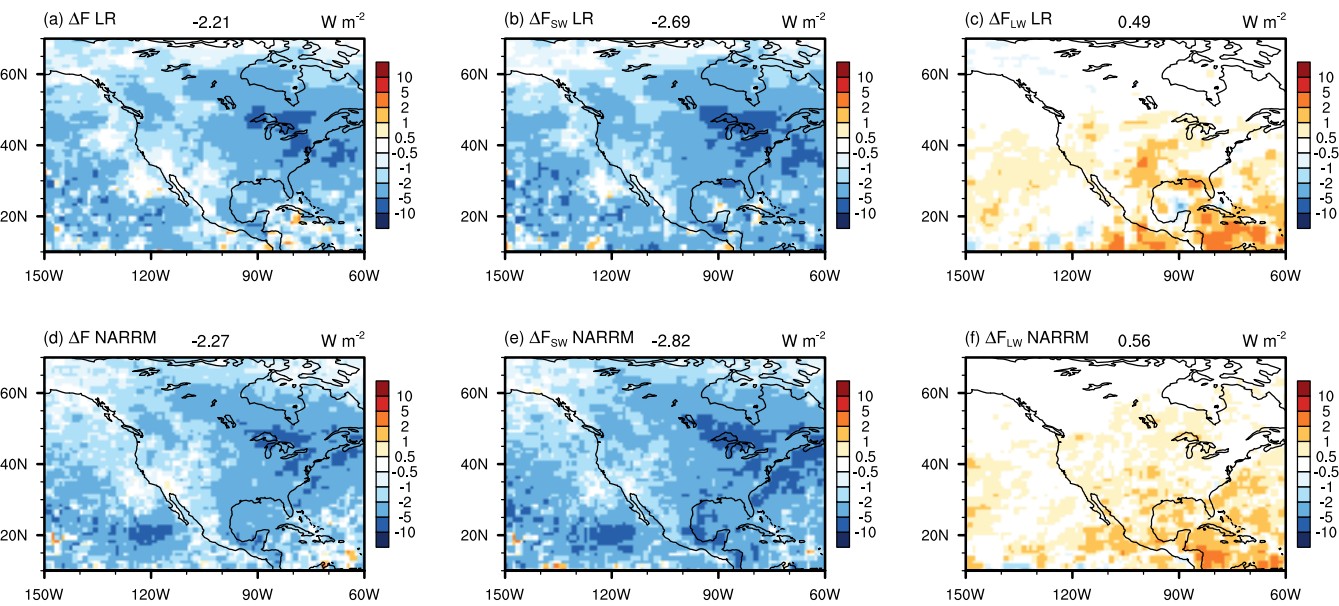

**Figure 17.** Anthropogenic aerosol effects simulated by the nudged LR and NARRM simulations.

over high latitudes (Bennartz et al., 2013; Hofer et al., 2019), which largely influences regional and global climate change (Tan et al., 2016; Tan and Storelvmo, 2019; Bjordal et al., 2020). Here we examine the impact of increased horizontal resolution on the simulated clouds and their radiative effects in the North polar region with the LR and NARRM historical simulations.

    Figure 18 compares the cloud cover between the E3SM CALIPSO (Cloud-Aerosol Lidar and Infrared Pathfinder Satellite Observation) simulator output and the GCM-Oriented CALIPSO Cloud Product (CALIPSO-GOCCP). Cloud cover and cloud
thermodynamic phase are diagnosed with the same algorithm in the CALIPSO simulator and CALIPSO-GOCCP data, facilitating consistent model-observation comparisons (Chepfer et al., 2008, 2010; Cesana and Chepfer, 2013). Observed total cloud cover is larger than 50% over the entire NA polar region and Greenland in the CALIPSO-GOCCP data. More extensive cloud cover is found over the North Atlantic. Strong land-ocean contrast is observed — liquid phase clouds dominate over the ocean while ice phase clouds prevail over the land such as Greenland and southern Alaska. Compared to CALIPSO-GOCCP,
LR overestimates total cloud cover at NA high latitudes, and underestimates at Greenland, particularly over the Baffin Bay and near the Greenland coast. The excessive modeled total cloud cover is primarily attributed to the positive biases in the liquid cloud with smaller contributions from the ice cloud over the western NA. In contrast, ice clouds are underestimated over Greenland and the northern Canada.

    Over land, NARRM displays improvements relative to LR in Alaska and Greenland for both cloud phases. For instance,
the ice cloud biases are significantly reduced in Alaska and western Canada (Fig. 18f, i), and the liquid cloud deficiencies over Alaska and Greenland are generally decreased (Fig. 18e, h). The better represented topography in NARRM (Fig. A1) is probably the key factor of these NARRM improvements. The impact of increased horizontal resolution on simulated cloud phase is also noted in the E3SMv1 model with the CALIPSO simulator (Zhang et al., 2019), where increased horizontal





resolution also slightly decreases simulated liquid and ice clouds at temperatures warmer than $-40°$C in the Arctic region. Over

ocean, NARRM moderately outperforms LR in representing liquid clouds over the North Atlantic to the west of Greenland. This is related to the warmer ($\sim$1.5°C) NARRM surface air temperature over the Labrador Sea. This warmer NARRM air temperature is consistent with the decreased sea ice concentration in that region (not shown), which is somewhat expected as an advantage of refining grids for both atmosphere and ocean/sea ice (see Fig. A2). Further process-level analysis is necessary to fully understand this LR-NARRM model behavior change and will be reported in separate papers.

Figure 19 compares the simulated SW and LW CRE with the CERES-EBAF Ed4.1 observations. Large negative SW CRE biases and positive LW CRE biases are shown over the NA in LR, which mainly result from the overestimated cloud cover. These biases are substantially reduced in NARRM primarily owing to the improved cloud cover (Fig. 18). As discussed by Golaz et al. (2022), sea ice concentration is largely overestimated over the North Atlantic in LR. The too large sea ice extent leads to weaker SW and LW CRE than observed at Labrador Sea. Compared to LR, the maximum positive bias in NARRM

sea ice extent in Labrador Sea is greatly alleviated. The improved representation of clear-sky radiative fluxes thus leads to a better CRE in NARRM.

    Figure 20 examines the impact of increased model resolution on surface energy fluxes. Compared with the ERA5 reanalysis data, sensible and latent heat fluxes are reasonably represented over the NA with subtle differences between LR and NARRM. However, LR surface energy fluxes have large biases in the North Atlantic and Greenland. For example, LR shows large

negative biases of sensible and latent heat fluxes over Labrador Sea and positive biases of sensible heat fluxes over Greenland. Large improvements are found in NARRM simulated sensible and latent heat fluxes relative to LR, with the most significant improvement in the North Atlantic. Although the negative biases exist, the magnitude is noticeably reduced in NARRM. As discussed before, the maximum sea ice concentration in Labrador Sea is less in NARRM than in LR, decreasing the positive sea ice biases (not shown). The better-resolved sea ice distribution and the associated sea surface temperature change are probably

the reason for the improved coupling between atmosphere and underlying surfaces in NARRM.

### 5.1.4 Extratropical cyclone

One of the primary motivations for pushing climate simulation resolution is to potentially better capture extremes. Extratropical cyclones (ETC) are a major weather extreme phenomenon in middle and high latitudes, bringing with them strong winds and precipitation that can exert substantial societal impacts along their pathways over days and over hundreds to thousands of

kilometers. Climate changes are likely to induce changes to the dynamical and physical characteristics of ETCs as well as their geospatial distribution (e.g. Bengtsson et al., 2006; Ulbrich et al., 2009). Projections of such future changes rely highly on numerical climate and ESM models. Their skills in simulating major weather systems like ETC have been carefully scrutinized by modeling centers and by the climate science community in conjunction with the major intercomparison campaigns (Greeves et al., 2007; Chang, 2013). While the conventional climate models with grid resolution around 100 km show reasonable skills

in producing ETC frequency and spatial track density, it has also been found that higher resolution models are better capable of capturing more intense ETCs (e.g. Jung et al., 2006), which is critical for using ESM to project future climates as growing evidence shows that global warming tends to shift the weather spectrum to the more extreme end (Melillo et al., 2014). Here,





**Figure 18.** Spatial distribution of annual mean total cloud cover (a), cloud cover in liquid phase (b), and ice phase (c) from the CALIPSO-GOCCP data. The cloud cover biases in LR (H1-5) and NARRM (H1-5) historical simulations (1985–2014) are shown in (d–f) and (g–i), respectively. Simulated cloud cover and cloud thermodynamic phase are derived by the CALIPSO simulator. Climatology data of CALIPSO-GOCCP version 3.1.2 (Chepfer et al., 2010) from 2006–2018 are used in the model evaluation.



**Figure 19.** Spatial distribution of observed shortwave cloud radiative effect (a), longwave cloud radiative effect (b), and simulated cloud radiative effect biases in LR (H1-5) (c, d) and NARRM (H1-5) (e, f) historical simulations (1985–2014). The observed cloud radiative effect is from the CERES-EBAF Ed4.1.







**Figure 20.** Spatial distribution of sensible heat flux (a) and latent heat flux (b) from ERA5 reanalysis data (Hersbach et al., 2020). Differences of sensible and latent heat fluxes between LR (H1-5), NARRM (H1-5) historical simulations (1985–2014) and ERA5 are shown in (c, d) and (e, f).



we will demonstrate the benefits of higher resolution in simulating ETCs in a regionally refined setting, by comparing the results from NARRM with those from LR simulations against the ETC activities derived from the ERA5 reanalysis.

The ETC tracks and statistics can be obtained using automated identification and tracking algorithms (e.g. Blender and Schubert, 2000; Geng and Sugi, 2001; Bengtsson et al., 2006; Jung et al., 2006; Ullrich and Zarzycki, 2017; Ullrich et al., 2021). The objective identification and tracking also make it suitable to compare ETC activities and statistics derived from different data sources in particular for model evaluations. The algorithms usually identify and track the spatial features of a meteorological variable, such as mean sea-level pressure (MSLP) or 850 hPa vorticity, that can characterize the structure of

cyclones and their movements. In this work, we use a community feature detection and tracking framework, TempestExtremes (Ullrich and Zarzycki, 2017; Ullrich et al., 2021) to derive ETC activities from 6-hourly MSLP data during the period of 1985–2014 from the E3SM simulations and the ERA5 reanalysis. Considering higher-resolution data can more accurately identify the storms and their tracks (Blender and Schubert, 2000; Geng and Sugi, 2001), all the model and reanalysis data are placed on 1°x1° grids to feed the tracking software. The algorithm takes two steps. First, a candidate cyclone is detected when a

minimum MSLP feature is enclosed by a contour of 200 Pa interval within 6 degrees of the center. Candidates within 6 degrees of one another are merged, with the lower center pressure taking precedence. The candidates are then stitched together to define the tracks if the features persist for at least 60 hours with a maximum gap of at most 18 hours. From the start to the end, a candidate cyclone must travel at least 12-degree great circle distance to qualify as an ETC.

Over the NARRM high-resolution domain, ETCs are most active during winter. Figure 21 shows the mean DJF track density

for the models and the analysis derived by casting the computed ETC track data onto 5°x5° grids. The tracks are mostly concentrated over the Northeast Pacific and Northwest Atlantic that form the well-known storm tracks. Both LR and NARRM simulations capture these main features to a large extent. There are, however, notable differences between LR and NARRM over these oceanic storm tracks. The track densities are clearly underestimated in the LR simulations inside the refined region, except for the Atlantic storm track in the coupled mode. NARRM clearly produces higher ETC track density than does LR

for both sections of the oceanic storm tracks, and mostly agrees better with the ERA5 data, although in the coupled mode, the track density tends to be overestimated. The shape and orientation of both the Pacific and Atlantic storm tracks are much better produced by NARRM in the coupled mode. Given the significant differences from both coupled LR (Fig. 21d) and uncoupled NARRM (Fig. 21c), it is reasonable to believe that the better captured storm track shapes in the coupled NARRM are due to interactions with the refined ocean (Fig. A2). Several secondary centers of active ETCs in the ERA5 over land are also

reproduced by the models, including the active regions over the Great Lakes and the Hudson Bay, though the densities are overestimated in the models, more so in the NARRM. It is worthwhile to mention that the NARRM simulations are able to produce the chain of secondary centers to the east of the Rocky Mountains that are also present in the ERA5 reanalysis but are largely missing in the LR simulations. This is presumably due to the NARRM's better resolved mountainous terrain features, a benefit by design.

The benefit of grid refinement can be further seen in Fig. 22, which shows the histograms of the ETC as a function of the minimum center pressure and the maximum deepening rate during its lifetime. All events within the refined region bounded by the black dashed lines as shown in Fig. 21 are used to compute these statistics. The maximum deepening rate is defined as





the maximum 6-hourly center pressure drop, normalized by $\sin(\varphi_{\mathrm{ref}})/\sin(\varphi)$, with $\varphi$ being the latitude and $\varphi_{\mathrm{ref}}$ the reference latitude at 45 degrees (see also in Jung et al., 2006). Clearly the NARRM very closely reproduces the number of intense

cyclones (minimum center MSLP < 960 hPa), while not surprisingly the LR model underproduces. These are true in coupled and uncoupled modes. Both LR and NARRM simulations overestimate the number of weaker ETCs. On a similar note, the observed number of rapid-growth cyclones are closely reproduced by the NARRM simulations, while clearly underestimated by a large margin in the LR simulations.

### 5.1.5 Cloud feedback

Since NARRM improves clouds and cloud radaitive forcing relative to LR over NA, we further examine the regional climate feedbacks over this region in Fig. 23. Relative to the global-mean value, the total climate feedback over NA is more negative from NARRM than from LR. This mainly results from the more negative Planck feedback and less positive SW cloud feedbacks. The more negative Planck feedback is related to the stronger surface warming over the Northeastern Pacific (not shown).

Figure 24 shows the spatial distribution of cloud feedback of LR and NARRM. Due to the different cloud types over land and ocean, we report the regional averages separately for land and ocean. Notably, the total land cloud feedback is 0.32 W/m$^2$/K smaller in NARRM (0.08 W/m$^2$/K) than in LR (0.39 W/m$^2$/K), which is dominated by the reduced SW cloud feedback over the Northeastern US (Fig. 24f). Further examination indicates this reduction is mainly related to the weaker reduction in low cloud fraction under warming in NARRM. Over the oceanic regions, NARRM presents a stronger SW cloud feedabck and a

weaker LW cloud feedback over the marine low cloud regime, leading to a small change of total cloud feedback. Across the CSS/WGNE Pacific Cross-Section Intercomparison (GPCI) transect (Teixeira et al., 2011), NARRM tends to show a weaker positive cloud feedback near the coast and more positive cloud feedback off the coast. These suggest that the regional refinement can significantly affect the regional cloud responses under warming and the predictability of regional climate.

### 5.2 Land

### 5.2.1 Hydrology

NARRM better captures spatial variability in land hydrologic processes, as indicated by two most important land hydrologic variables, total runoff (Fig. 25) and evapotranspiration (ET) (Fig. 26). For instance, in the pacific coastal regions, the Rocky Mountains block atmospheric moisture from ocean to inland areas and lead to two distinct hydrologic regimes: a wet regime in the mountain west and a dry regime in the mountain east. This abrupt spatial shift from a wet to dry hydrologic regime can be

clearly seen in the composite runoff map from the Global Runoff Data Center (GRDC) (Fekete et al., 2011), Fig. 25a, and the observed evapotranspiration map from the Moderate Resolution Imaging Spectroradiometer (MODIS) satellite observations (Running et al., 2017), Fig. 26a. Note that the GRDC runoff map is not completely based on the observational data since runoff measurements are not available at the regional or global scales due to technical and economic limitations. It is nevertheless a more realistic estimate than any model simulations because it was first generated with a monthly hydrologic model (hence



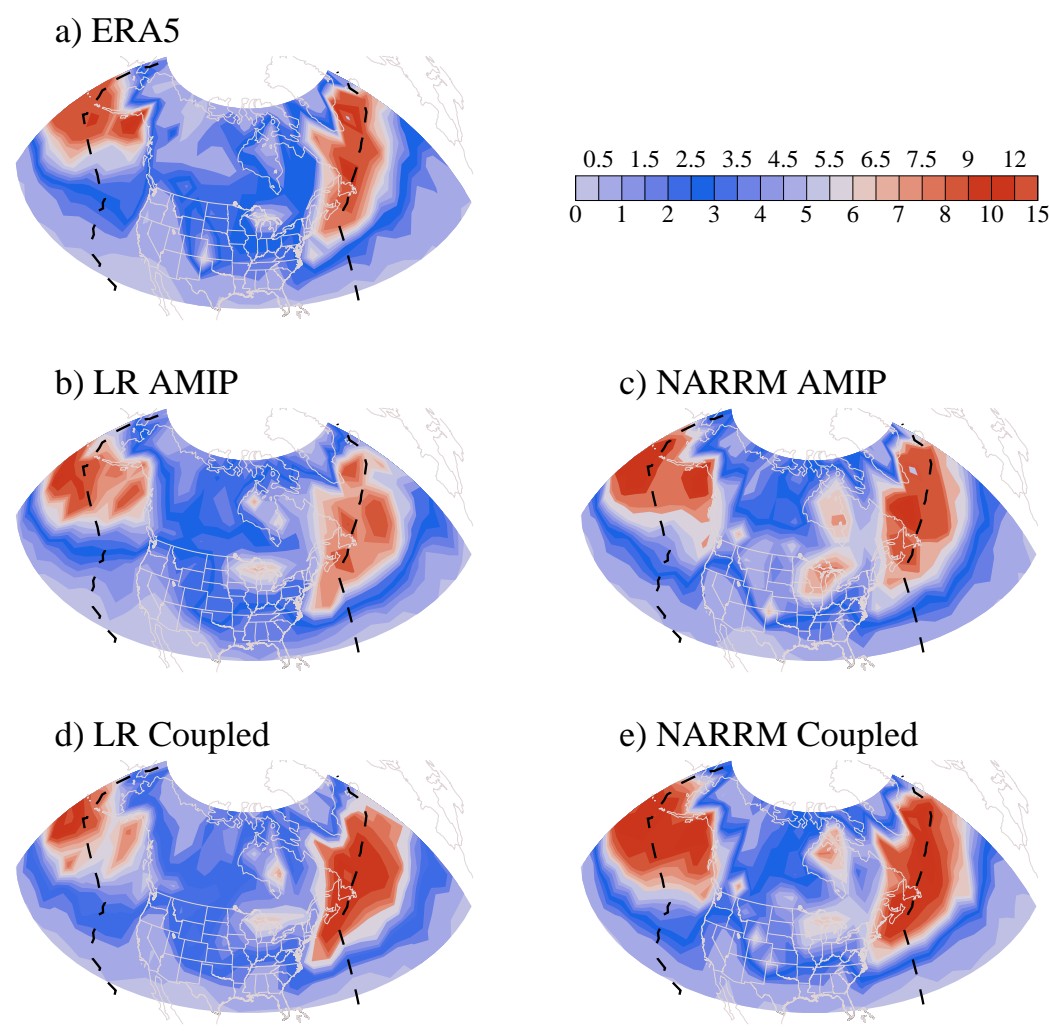

**Figure 21.** Mean extratropical cyclone track density in the DJF season between 1985 and 2014 from (a) ERA5, (b, c) AMIP, and (d, e) historical simulations of LR and NARRM. Black dashed lines denote the western and eastern boundary of the refined region.





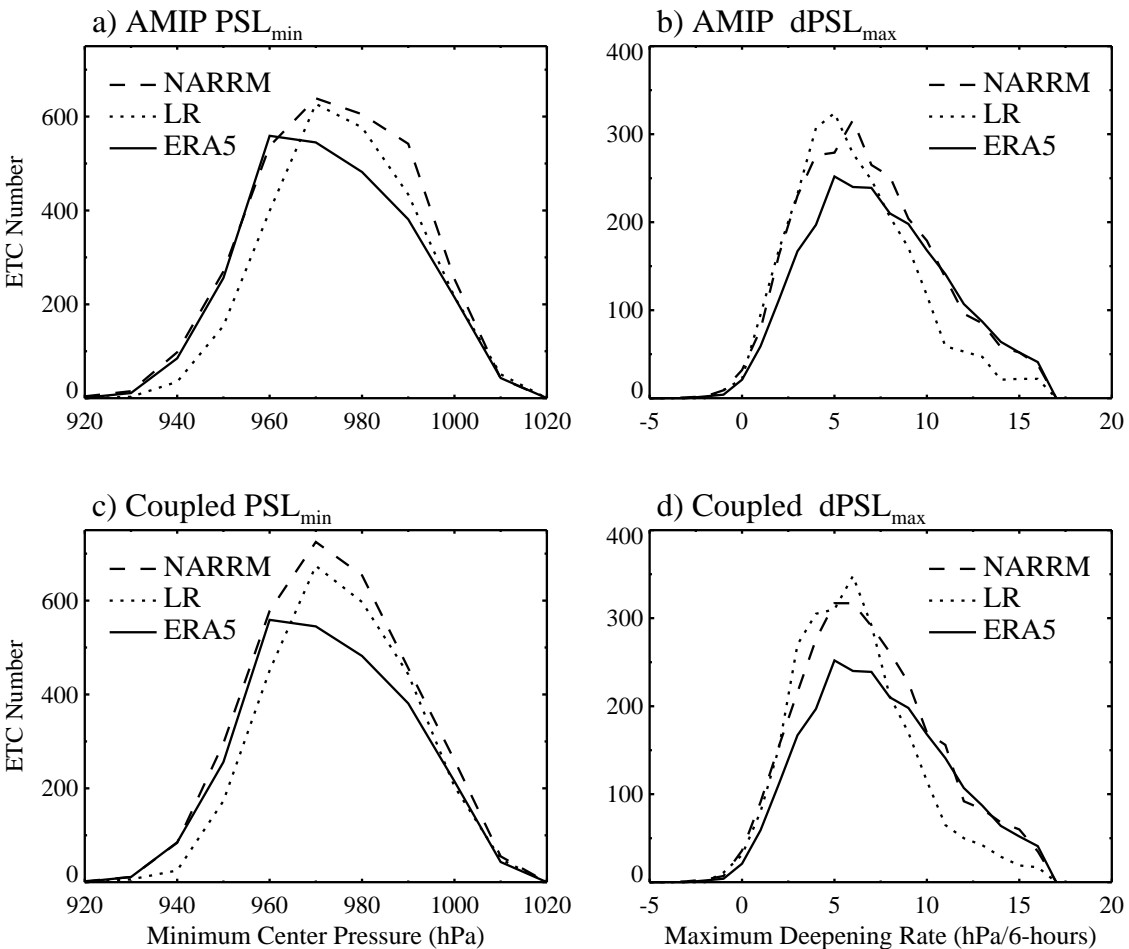

**Figure 22.** Histogram of extratropical cyclone minimum center sea level pressure and maximum 6-hour deepening rate in the DJF season between 1985 and 2014 (a, b) AMIP and (c, d) historical simulations. LR is shown in dotted, NARRM in dashed, and ERA5 in solid black lines.



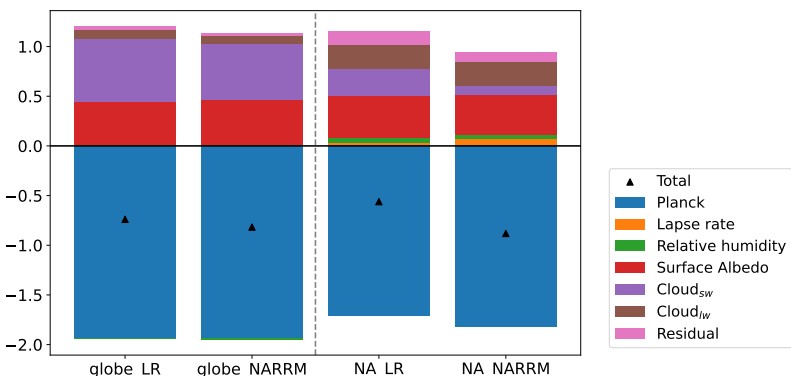

**Figure 23.** Global- and North America-mean climate feedbacks of LR and NARRM decomposed using radiative kernels (e.g., Soden et al. (2008)).

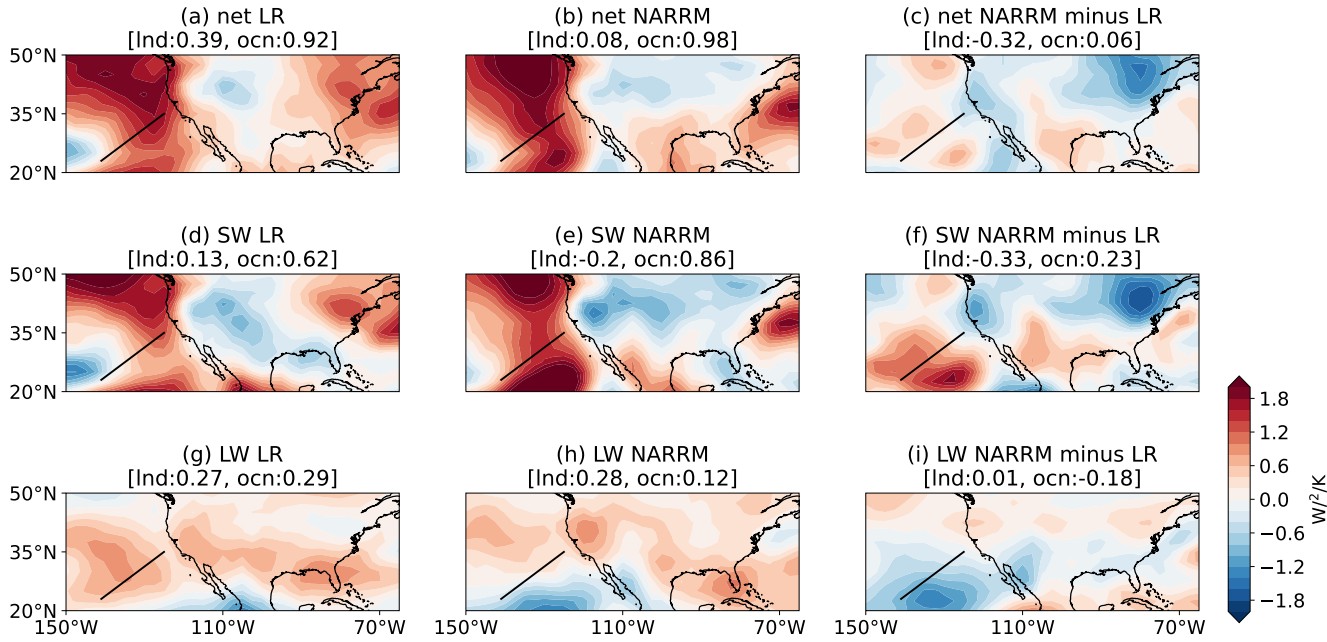

**Figure 24.** Spatial distribution of North America (NA) total (a-c), SW (d-f), and LW (g-i) feedbacks for LR, NARRM, and the difference between NARRM and LR. The GPCI transect (Teixeira et al., 2011) is denoted by the black line. The average values over land and ocean are labelled in the brackets.



producing spatiotemporal variability) and then bias-corrected against discharge measurements at thousands of river gauges
(Fekete et al., 2011). This abrupt shift of hydrologic regime around the Rocky Mountains, along with the other spatial variations,
is much better resolved in the NARRM simulation than LR, which is the case for both simulated runoff and evapotranspiration,
as shown in Figs. 25b, c and 26b, c. The NARRM simulated spatial patterns are thus more realistic than the LR ones over NA.
Over the remaining regions of the globe, the NARRM and LR simulated spatial patterns are quite similar to each other in terms
of both runoff and evapotranspiration (not shown).

The simulation biases in runoff and evapotranspiration are further examined in terms of absolute biases, i.e., absolute differ-
ence between the simulated and "benchmark" values. Here the GRDC runoff and MODIS ET data are used as the "benchmark"
data. Figures 25d and 26d show the maps of absolute bias difference, i.e., difference between the absolute biases in the LR
simulation and those in the NARRM simulation (former subtracting latter), for annual mean runoff and evapotranspiration,
respectively. For a specific grid cell in these two maps, a positive difference means the absolute bias in the LR simulation is
larger than that in the NARRM, and vice versa. It appears that there are more absolute biases in LR than NARRM over both
western and eastern U.S. Using the longitude 100°W as the divide, the average absolute bias differences (positive indicates
NARRM has less overall absolute bias than LR) are 22.8 mm/yr and 0.9 mm/yr over the western and eastern U.S., respectively,
for annual mean runoff, and are 21.6 mm/yr and 18.5 mm/yr over the western and eastern U.S., respectively, for annual mean
evapotranspiration. Comparing to LR, NARRM can thus help reduce simulation biases in hydrologic variables.

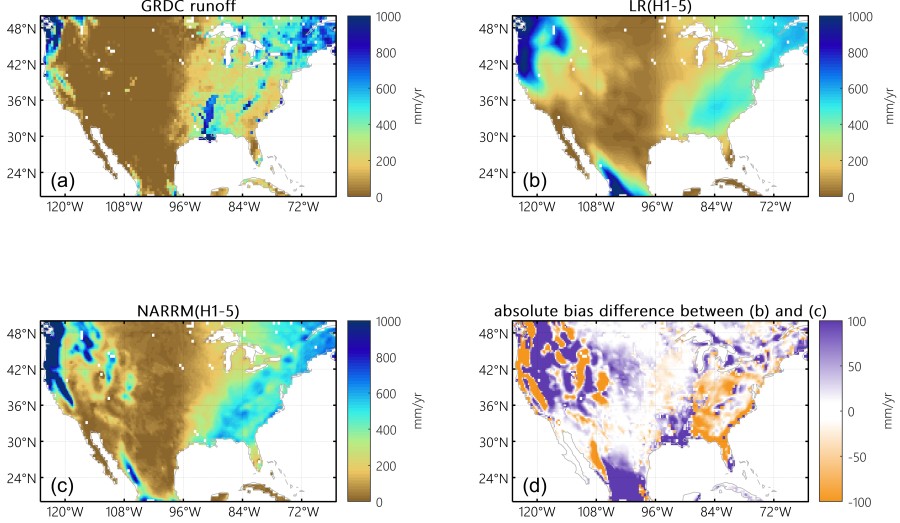

**Figure 25.** Annual mean runoff from (a) GRDC, (b) LR, (c) NARRM simulation, as well as (d) the differences of absolute biases between LR
and NARRM (former subtracting latter), a positive value suggests the absolute bias in the LR simulation is larger than that in the NARRM,
and vice versa.





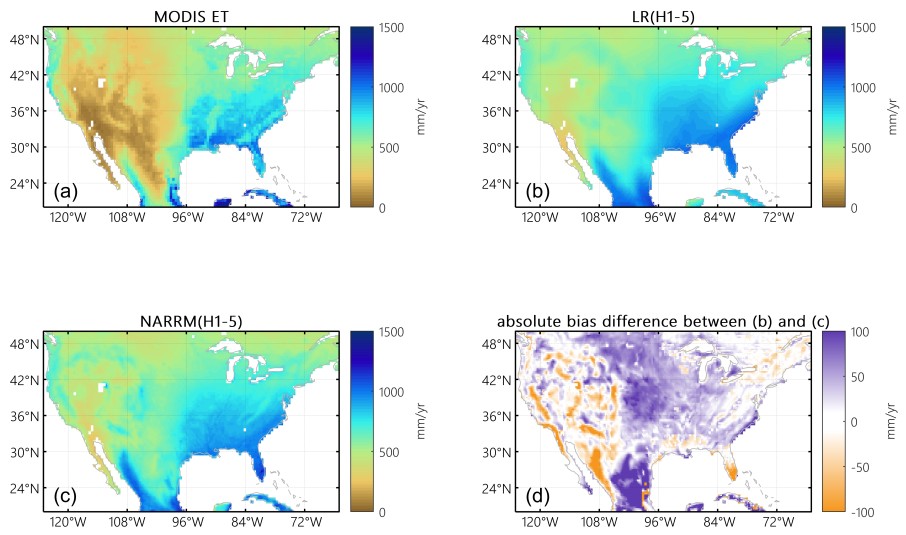

**Figure 26.** Annual mean evapotranspiration from (a) MODIS, (b) LR, (c) NARRM simulation, as well as (d) the differences of absolute biases between LR and NARRM (former subtracting latter), a positive value suggests the absolute bias in the LR simulation is larger than that in the NARRM, and vice versa.

### 5.2.2 Snowpack

Natural storage provided by mountain snowpack is central to water supply reliability in the western U.S. (Siirila-Woodburn et al., 2021). To evaluate model skill in representing this critical hydroclimate benchmark variable, intra-annual snowpack dynamics are evaluated using the methodology known as the snow water equivalent (SWE) triangle (Rhoades et al., 2018a, b).

The seven metrics that make up the SWE triangle attempt to distill management-relevant aspects of the accumulation and ablation of snowpack (e.g., peak water volume and snowmelt rate) for any arbitrary gridded SWE dataset. Five HUC2 basins of the mountainous western U.S. are used to derive five-member ensemble- and basin-average evaluations of LR and NARRM fully-coupled historical simulations and are compared with ERA5. All datasets are bi-linearly regridded using ESMF to 0.25° resolution prior to masking and computing the basin-average SWE triangle metrics.

NARRM provides enhanced winter (DJF) climatological representation of the spatial variability of SWE across the CONUS relative to LR (Fig. 27a,b). This is seen through higher SWE magnitudes and more granular spatial structures in NARRM compared with LR, particularly in coastal mountain ranges such as the Cascades and Sierra Nevada, and corroborates a long-history of ESM studies that highlight the critical importance of horizontal resolution (≤0.25°) in properly representing the mountainous hydrologic cycle (Demory et al., 2014; Rhoades et al., 2017; Kapnick et al., 2018; Palazzi et al., 2019; Bambach

et al., 2021; Rhoades et al., 2021). As shown through the more granular intra-seasonal perspective of the SWE triangle met-



rics, certain aspects in the snowpack dynamics are improved with NARRM (e.g., peak water volume), namely in the Pacific Northwest and California (Fig. 27c). With that said, some E3SM SWE biases are not ameliorated with horizontal resolution and may arise due to the combination of higher winter-season precipitation (Fig. 11) and general cool-bias (Fig. 8) in both the LR and NARRM fully-coupled historical simulations.

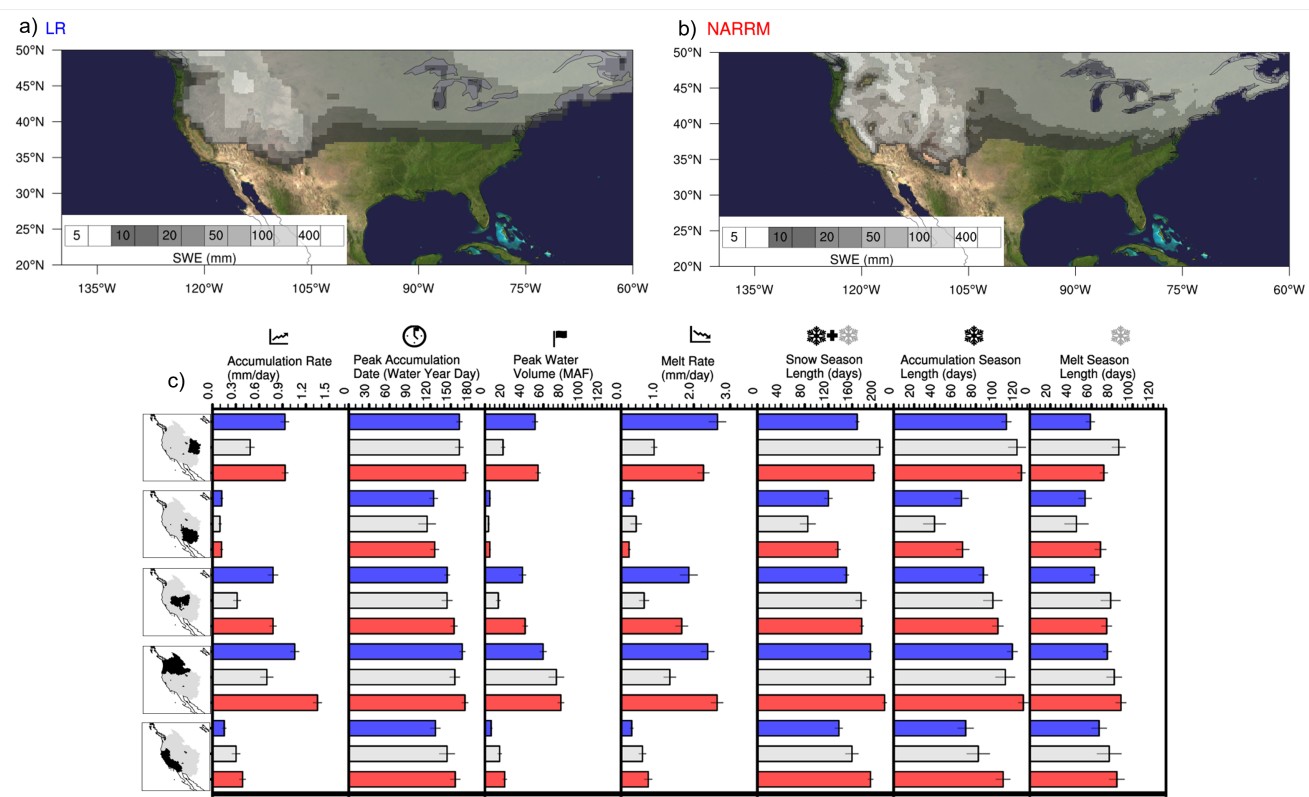

**Figure 27.** Climatological DJF average snow water equivalent (SWE) as simulated by E3SMv2 at (a) low-resolution (LR) and (b) with the North American Regionally Refined Mesh (NARRM) over the 1985–2014 period. (c) LR (blue) and NARRM (red) SWE triangle metrics for five HUC2 basins within the mountainous western U.S. compared with ERA5 (gray). Black-bars at the end of each histogram represent the mean 95% confidence intervals.

## 5.3 Land-atmosphere coupling


Accurate representation of the interactive processes between the land surface, planetary boundary layer (PBL), and clouds and precipitation is an ongoing challenge for current state-of-art climate models. In this Section, we assess the land-atmosphere (L-A) coupling in LR (H1-5), LR (A1-3), NARRM (H1-5), and NARRM (A1-3) using the 9-yr warm-season observations at the Atmospheric Radiation Measurement (ARM) Southern Great Plains (SGP) site, following Tao et al. (2021). Before the detailed analysis on L-A coupling, we first examine the seasonal variations of daytime mean surface heat fluxes from May to August






during 2004-2012. As shown in Fig. 28, the evaporative fraction (EF) is in general underestimated in model simulations except for LR (H1-5). The much lower simulated EF compared with the ARM observations is mainly attributed to a large negative bias in surface latent heat fluxes (LH). Different from the other simulations, the surface sensible heat fluxes (SH) is significantly underestimated in LR (H1-5) from May to early July. As a result, the simulated daytime mean EF is higher than that from

the observations. Overall, the surface state and fluxes are relatively better reproduced in the historical runs than in the AMIP runs, where both LR (A1-3) and NARRM (A1-3) show a significant negative bias in LH and EF persisting since July. This is surprising and requires further analyses, which is beyond the scope of this paper. In the following, we focus on two local convective regimes and diagnose model behaviors using the Local Coupling metrics (Santanello et al., 2018).

During the selected 9-year period, 165 and 154 clear-sky days are classified from LR (A1-3) and NARRM (A1-3), respec-

tively (Table A1). This is double the 66 clear-sky days identified from ARM observations. However, the occurrence frequency of shallow cumulus (ShCu) days is much lower in these AMIP runs compared with that observed. Only 6 and 5 ShCu days are identified in LR (A1-5) and NARRM (A1-5), respectively (not shown). For the historical runs, the number of selected clear-sky days from both LR (H1-5) and NARRM (H1-5) are comparable to that observed but the occurrence frequency of ShCu days is still low. As we are targeting for a statistical and climatological comparison between the long-term ARM data and climate

model simulations, we extend the analysis period to 1980-2012 for model simulations on ShCu days due to the limited sample size between 2004 and 2012.

Figure 29 shows the composite clear-sky day mixing diagrams (Santanello et al., 2009), which relates the conservative variables, potential temperature ($\theta$) and total water specific humidity (q), to the water and energy budgets and the growth of planetary boundary layer (PBL). The coevolution of Lvq and Cp$\theta$ (0730 to 1730 LST) is decomposed by vector components

that represent the integrated fluxes of heat and moisture from the land surface ($V_{sfc}$), the advection ($V_{adv}$) and the entrainment at the PBL top ($V_{ent}$ as a residual). Six metrics are derived from these diagrams and summarized in Table 5. In general, although the differences among various model simulations are minor, several common model biases are noted when compared with the ARM observations. For example, the model simulated clear-sky days are featured with too warm and too dry conditions in the early morning (0730 LST). The $\theta_{sfc}$ in the two AMIP runs and historical runs, both LR and NARRM, are about three and

two times of that observed, respectively. The high $\theta_{sfc}$ indicates that more energy at the surface goes to heating rather than moistening. Moreover, the $E_{LH}$ is significantly overestimated in the model simulations, which is about five (three) times of that observed in the two AMIP runs (historical runs). The much higher simulated $E_{LH}$ suggests that the entrainment heating and drying dominate the surface fluxes on the simulated clear-sky days, which supports rapid and deep PBL growth in models. Different from the observations, the simulated advection tends to cool and dry the mixed layer, but the overall impact is much

smaller compared with those from the surface and entrainment.

Figure 30 shows the daytime evolution composites of PBL, lifting condensation level (LCL), and LCL deficit (PBL top height minus LCL) on clear-sky and ShCu days. Both PBL and LCL on model simulated clear-sky days are much higher than that on the ARM observed clear-sky days. The model behaviors are in general consistent among different simulations, except that the bias of LCL is significantly lower in LR (H1-5) compared with the others. In models, the PBL grows rapidly after

the sunrise on clear-sky days, corresponding to the large warm and dry air entrainment that dominate the PBL budget (Fig.





29). But the too warm and too dry early-morning surface conditions lead to an even higher LCL on model simulated clear-sky days. The PBL never reaches the LCL, with a negative LCL deficit throughout the day, which supports clear skies. The diurnal evolution of LCL on the ARM-observed ShCu days is similar to that on clear-sky days but the development of PBL is much more vigorous. As a result, the PBL is deep enough to touch the LCL for cloud formation around noon. Different from that

observed, the daytime evolution of PBL is much weaker on ShCu days than that on clear-sky days in all model simulations. However, the decrease in LCL from clear-sky days to ShCu days is even greater, where the growth of PBL is high enough to touch the LCL for cloud formation. Note that the models simulate a positive LCL deficit at around 0900 LST, a few hours earlier than that in the observations. To summarize, ShCu forms as a result of strong surface SH fluxes that drives the rapid development of PBL in observations while in models, ShCu results from a relatively more humid lower troposphere that leads

to a lowered LCL. Differences among various model simulations are pretty minor.

**Table 5.** The surface ($\beta_{sfc}$) and entrainment ($\beta_{ent}$) Bowen ratios, the entrainment ratio of heat ($E_{SH}$) and moisture ($E_{LH}$), and the advective flux ratio of heat ($A_{SH}$) and moisture ($A_{LH}$) from the ARM observations, LR (A1-3), LR (H1-5), NARRM (A1-3), and NARRM (H1-5) on clear-sky days. The flux values (W/m$^2$) are derived using the mixing diagram theory and surface, advection, and entrainment flux vectors depicted in Figure 29.

| Metrics | Obs. | LR (A1-3) | LR (H1-5) | NARRM (A1-3) | NARRM (H1-5) |
|---|---|---|---|---|---|
| $\beta_{sfc} = \text{SH}_{sfc}/\text{LH}_{sfc}$ | 0.70 | 2.28 | 1.20 | 2.09 | 1.65 |
| $\beta_{ent} = \text{SH}_{ent}/\text{LH}_{ent}$ | -1.13 | -0.68 | -0.60 | -0.54 | -0.61 |
| $E_{SH} = \text{SH}_{ent}/\text{SH}_{sfc}$ | 1.79 | 1.57 | 1.72 | 1.35 | 1.37 |
| $E_{LH} = \text{LH}_{ent}/\text{LH}_{sfc}$ | -1.11 | -5.31 | -3.44 | -5.24 | -3.72 |
| $A_{SH} = \text{SH}_{adv}/(\text{SH}_{sfc}+\text{SH}_{ent})$ | 0.04 | -0.18 | -0.12 | -0.09 | -0.07 |
| $A_{LH} = \text{LH}_{adv}/(\text{LH}_{sfc}+\text{LH}_{ent})$ | 3.97 | -0.21 | -0.25 | -0.22 | -0.22 |

## 5.4   River

Streamflow simulations are typically affected by multiple sources of uncertainties, such as the biases in the simulated precipitation and runoff, the uncertainties in the river model parameters (e.g., river network topology, channel geometry, Manning's roughness coefficients), and water demand data (Li et al., 2013, 2015a, b; Zhou et al., 2020). For river network topology, the

half-degree-resolution and 1/8th-degree-resolution river network data are used for the LR and NARRM simulations, respectively, as shown in Figs. 31a,b. It is expected that a higher resolution river network data can represent rivers more smoothly, hence more realistically. Another benefit of higher resolution river network data is to enable more extensive streamflow validation. Terrestrial water fluxes, particularly surface runoff and streamflow, are dominated by gravity and controlled by topography, hence mostly follow irregular watershed boundaries. In most land surface and ESMs, including E3SM, regular lat/lon grids are

used to resolve spatial heterogeneity for both runoff and river processes to be compatible with the other land and atmospheric components (Lawrence et al., 2019; Golaz et al., 2022). Both the magnitude and timing of streamflow at each river gauge are dominated by the corresponding upstream drainage area. Streamflow simulations are thus largely affected by the discrepancies



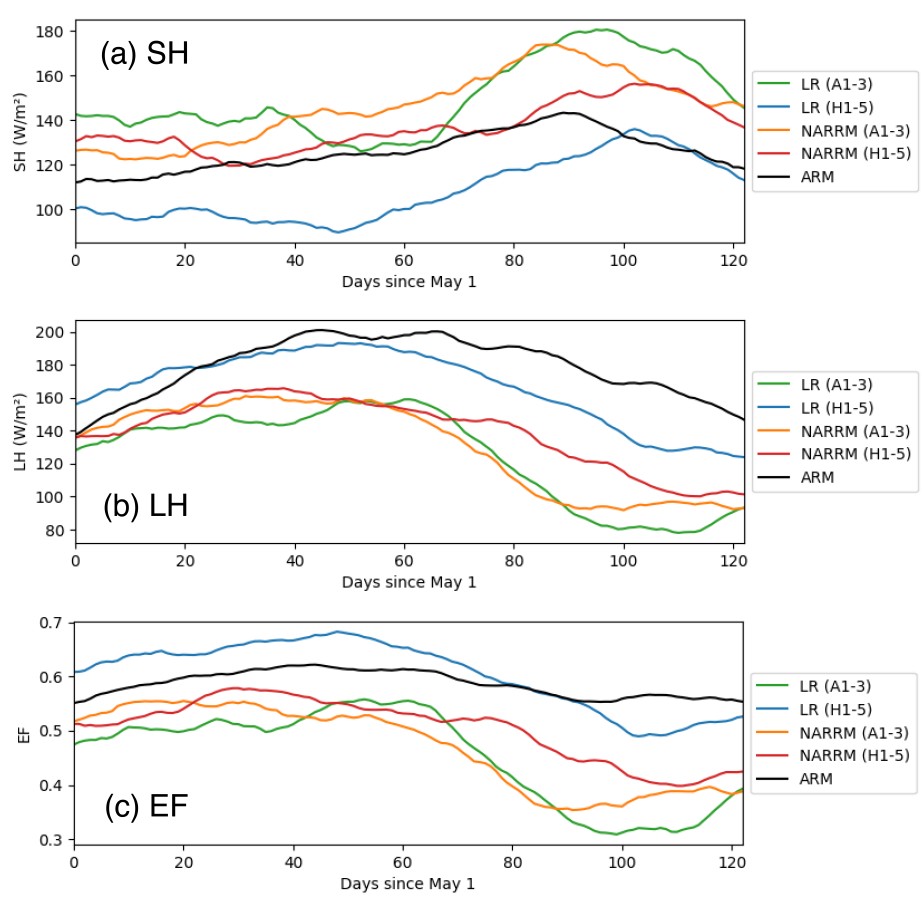

**Figure 28.** The seasonal variation of 2004-2012 daytime mean (0600-1800 LST): (a) surface sensible heat flux (SH), (b) surface latent heat flux (LH), and (c) surface evaporative fraction (EF, defined as [LH/(LH+SH)]) from ARM observations (black), LR (A1-3) (green), LR (H1-5) (blue), NARRM (A1-3) (orange), and NARRM (H1-5). A moving average of 30 days is applied to smooth out short-term fluctuations and highlight longer-term trends.





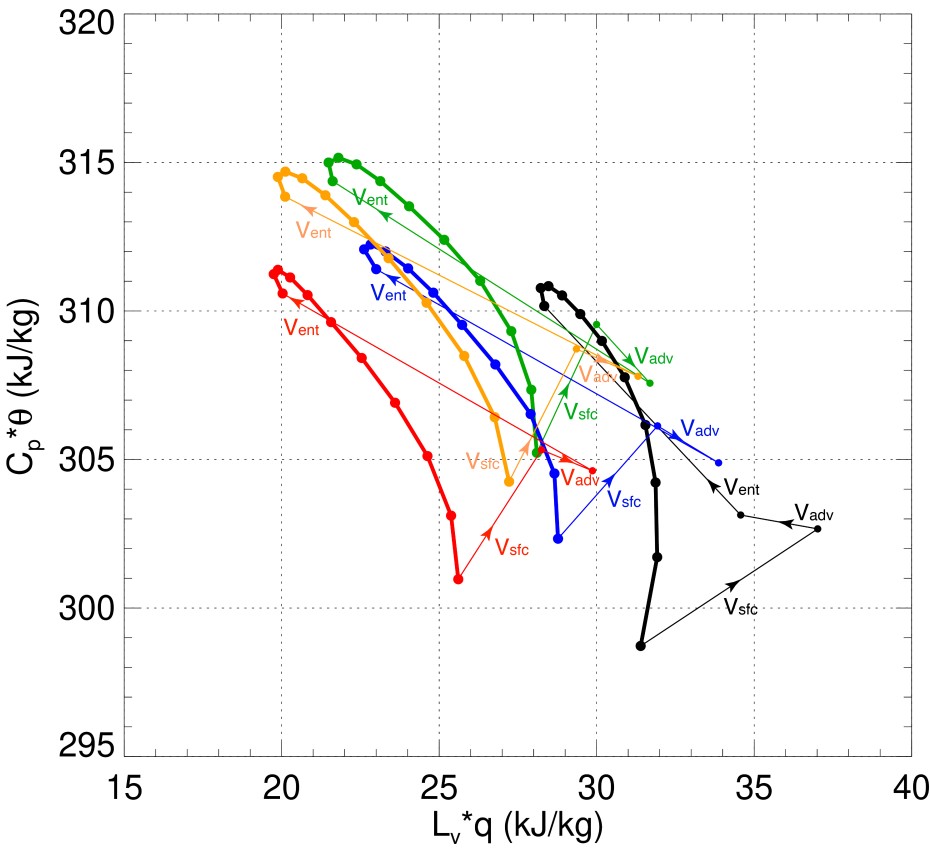

**Figure 29.** Clear-sky day mixing diagram of the PBL conservative variables, Lvq versus Cpθ, during the daytime evolution from ARM observations (black), LR (A1-3) (green), LR (H1-5) (blue), NARRM (A1-3) (orange), and NARRM (H1-5) (red). Dots denote the composite hourly means from 0730 to 1730 local time. The text annotations depict the vector component contributions from surface (Vsfc), advection (Vadv) and entrainment fluxes (Vent) to the evolution.



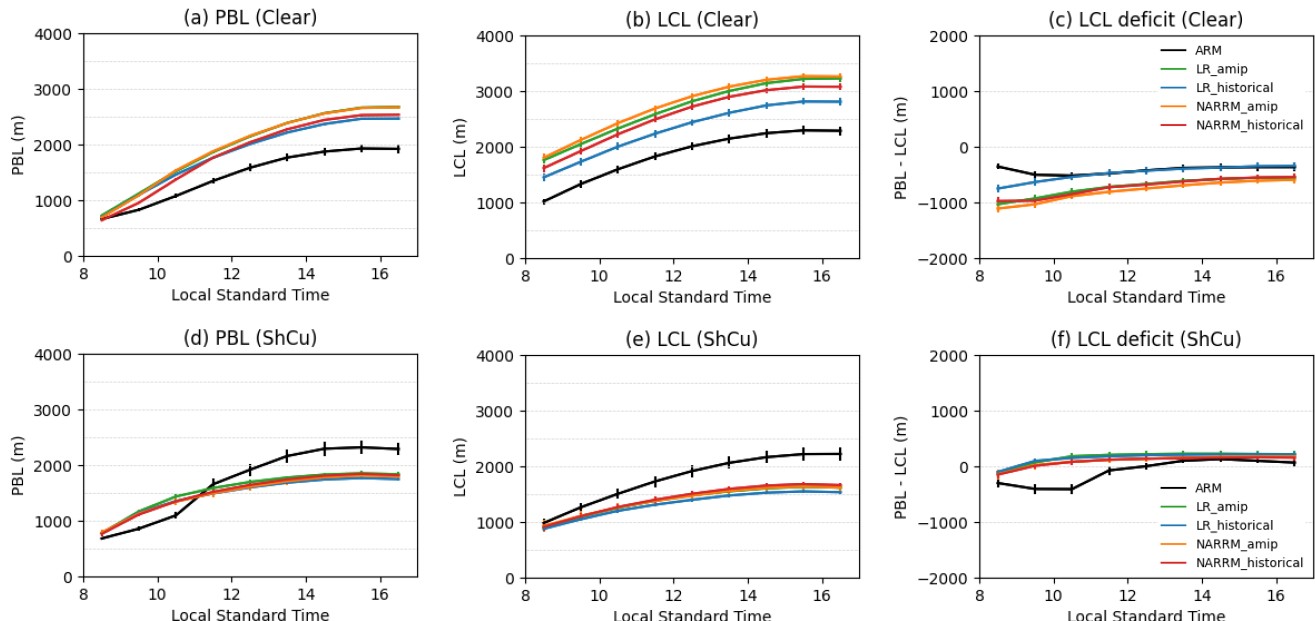

**Figure 30.** Composite daytime evolution of (a) PBL, (b) LCL, and (c) LCL deficit (PBL minus LCL) from ARM observations (black), LR (A1-3) (green), LR (H1-5) (blue), NARRM (A1-3) (orange), and NARRM (H1-5) (red) on clear-sky days. (d–f) are as (a–c) but on shallow cumulus days.

between the watershed boundaries and regular lat/lon grids. These discrepancies can be significantly reduced with higher resolution river network data. For example, in this study a 10% discrepancy threshold is used to select the river gauges for validating
streamflow simulations, i.e., the relative difference between the real upstream drainage area of a river gauge and that estimated from a lat/lon grid-based river network should not exceed 10%. Over the NA domain, 615 river gauges satisfy the requirement for the half-degree-resolution river network, whilst 2924 river gauges satisfy the requirement for the 1/8th-degree-resolution river network. There are 563 river gauges simultaneously satisfy the requirement for both resolutions.

     Figure 31c shows the comparison between the annual mean observed and simulated streamflow over these 563 river gauges.
Overall, both simulations produce the long-term average streamflow reasonably well across these gauges. NARRM performs noticeably better (closer to the red 1:1 line) for the top four gauges with largest discharges. An additional analysis (figure not shown) indicates that LR produces greater absolute bias than NARRM in 330 out of 563 gauges (about 60%) in the streamflow simulation. Interestingly, it appears that the over- and underestimation of JJA and DJF streamflow are concentrated in the western and eastern U.S., respectively, for both LR and NARRM, as shown in Figs. 32a–d. Figures 32e,f display the difference
in absolute biases between LR and NARRM (former subtracting latter) at individual river gauges for the JJA and DJF seasons. Positive difference (indicating greater bias in LR than in NARRM, purple color) dominates over most gauges in eastern US during JJA and over the CONUS during DJF. Taken together, Figs. 31 and 32 suggest an overall better performance of NARRM, despite all the uncertainties.





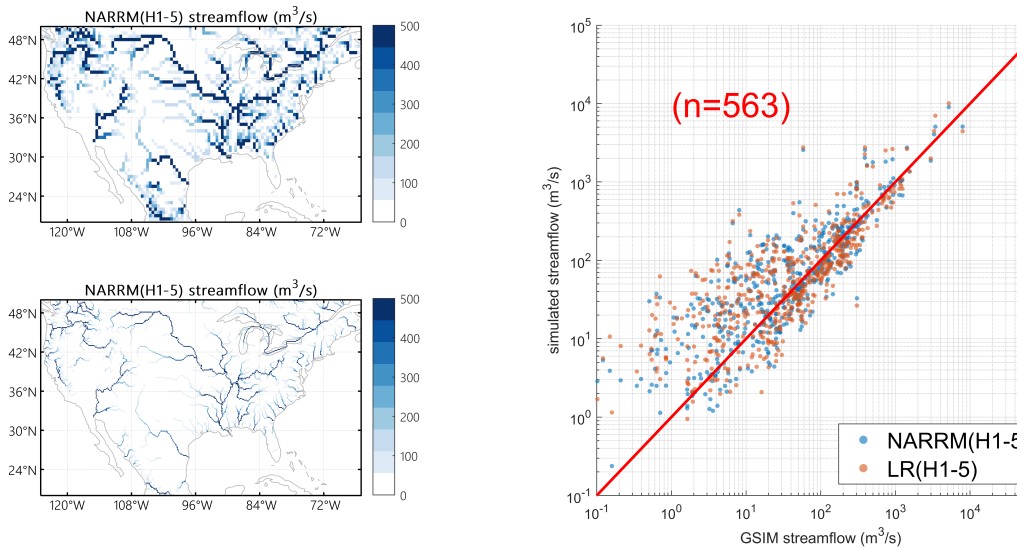

**Figure 31.** Simulated annual mean streamflow at 563 river gagues in NA compared against The Global Streamflow Indices and Metadata (GSIM) database. (a) River network used in LR, demonstrated by the mean annual discharge; (b) river network used in NARRM, demonstrated by the mean annual discharge; (c) simulated annual mean streamflow against observed for LR and NARRM.

## 6 Conclusions and discussions

A primary Earth system model (ESM) advancement is to represent the spatially continuous world more realistically on discretized grids, which often requires constantly increasing the finest scale of explicitly resolved processes within the computational limit. Before uniformly high-resolution global models solve their severe computational challenge for climate simulation campaigns, the multiresolution ESM (e.g., regionally refined model (RRM)) is a natural alternative for these campaigns. Nevertheless, it has been over a decade since such a multiresolution method (e.g., Ringler et al., 2008) was proposed.

To our knowledge, this is the first study with a global ESM that has accomplished the CMIP6 climate simulation campaign with a fully coupled RRM configuration — a potentially significant step in the long journey of improving the explicitly resolved resolution of climate simulations. The key to this success is the application of the hybrid timestep strategy (i.e., merging high-resolution dynamics timestep with the low-resolution physics timestep) in the atmosphere model, which bypasses the persistent poor scale-aware problem of atmospheric physics in a multi-scale framework (e.g., RRM). The powerful aspect of RRM is that

it typically only costs ~10–20% of the globally uniform high-resolution model, substantially reducing the computational burden of production simulations. This is particularly important for high-resolution ensemble simulations, which are necessary to account for the internal variability of the climate system, but whose cost would otherwise be prohibitive. On the global scale, we show that NARRM well reproduces the LR climate. Within the high-resolution domain (i.e., North America), NARRM displays more improvements than deteriorations relative to LR. Furthermore, some of the NARRM improvements (e.g., marine

shallow cumulus clouds at California and mixed-phase clouds near Arctic) are likely attributable to the better captured coupling



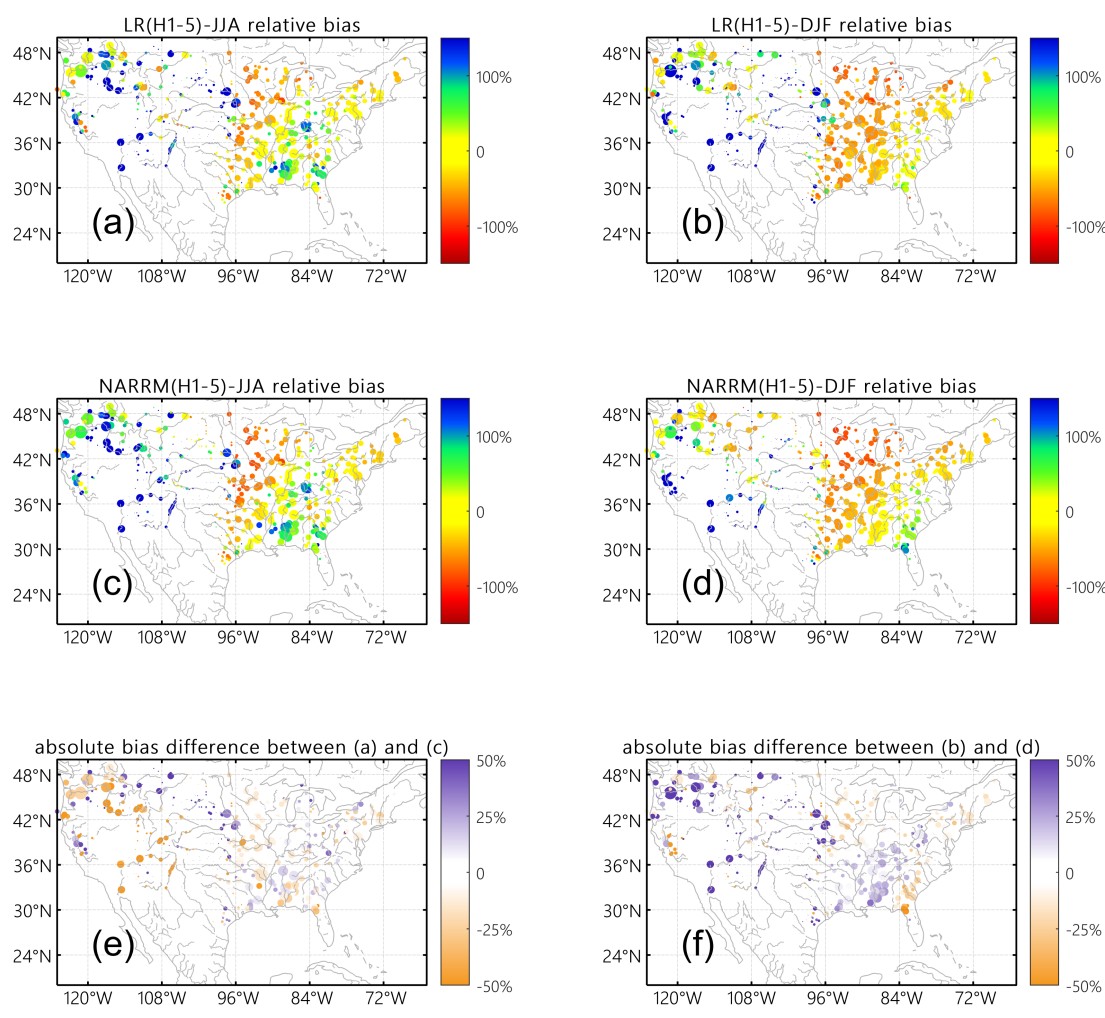

**Figure 32.** Simulated JJA (left panel) and DJF (right panel) mean streamflow at CONUS river gagues compared against The Global Streamflow Indices and Metadata (GSIM) database. (a) and (b), relative bias of LR; (c) and (d), relative bias of NARRM; (e) the difference of absolute relative bias between LR (absolute value of a) and NARRM (absolute value of c) for the JJA season, positive value (purple) indicates LR has greater absolute bias than NARRM; (f) same as e but for the DJF season





processes, highlighting the strength of refining multiple components over a single component. The main detailed findings are as follows.

– The new dry baroclinic idealized test (Hughes and Jablonowski, 2022) allows us to test the NARRM grid with the stand-alone atmospheric dynamical core and confirms that the NARRM mesh is numerically stable, and the results are
reasonable compared to the LR and HR grids (Fig. 4).

– By employing the EAM hybrid timestep method, NARRM, without retuning physics parameters, successfully matches the global climate (including climatology and climate sensitivity and feedback) simulated by LR (Figs. 7, 8, 9, 23).

– Within the high-resolution region over the CONUS, precipitation and clouds are largely improved in NARRM compared to LR (Figs. 10ac, 11, 12) due to the better topography in NARRM (Fig. A1) and/or reduced sea surface temperature
biases.

– Refining the atmospheric grid spacing from 100 km to 25 km is not adequate to improve the diurnal propagation of organized MCSs at the CONUS (Fig. 13).

– NARRM retains the LR performance of aerosol simulations on the global scale and regional mean basis, without re-tuning of the scale-dependent aerosol emissions. Over the refined mesh, NARRM improves the simulated aerosol spatial
variability and predictions of extreme polluted cases (e.g., the upper tail of AOD distribution) (Fig. 16). On the other hand, the refined grid resolution does not eliminate the high biases in aerosol loadings and effective radiative forcing, inherited from the LR model.

– In the North polar region, NARRM generally simulates better cloud covers than LR for both liquid and ice phase clouds. Over land (e.g., Alaska and Greenland), this improvement is likely related to topography, whereas over ocean, it is
attributed to air-sea interactions.

– While both the LR and NARRM simulations are able to capture to a large extent the spatial and statistical distributions of the observed extratropical cyclone (ETC) activities, NARRM shows a particularly better skill in simulating the ETC activities along the oceanic storm tracks and over the mountain range to the east of the Rocky Mountains (Fig. 21). NARRM in coupled mode outperforms all other configurations (LR and uncoupled NARRM) in simulating the shape
and orientation of the oceanic storm tracks within the NARRM high-resolution domain, due to the coupling with the refined ocean surrounding North America. NARRM in general produces more ETCs than LR, and overestimates the total number of cyclones compared to the ERA5 reanalysis. But probably more importantly, for intense cyclones or rapidly developing ones, the NARRM simulations are in close agreement with the observations while the LR simulations mismatch by a significant margin (Fig. 22).

– NARRM produces a comparable global-mean cloud feedback to LR but a less positive cloud feedback over the CONUS (Fig. 23). The reduction in cloud feedback there mainly relates to the shortwave component. The total cloud feedbacks





over coastal California does not change much due to the compensation between shortwave and longwave components (Fig. 24).

– NARRM appears to better represent the spatial variability in land hydrologic processes by resolving the land features more realistically over the western US (Figs. 25, 26). With higher grid resolution, NARRM can better capture surface topography which dominates surface water flows across hillslopes and through rivers, hence not only improving the river model performance but also providing more precise river gauge geo-referencing information for streamflow validations (Figs. 31, 32).

– NARRM provides enhanced winter (DJF) climatological representation of the spatial variability of snow water equivalent (SWE) across the CONUS relative to LR (Fig. 27a,b) as a result of higher SWE magnitudes and more granular spatial structures in NARRM. Certain biases (e.g., peak water volume) of snowpack are reduced in NARRM compared with LR (Fig. 27c).

   – Over the ARM SGP site during warm seasons, the surface conditions are warm and dry on the model simulated clear-sky days with overestimation in both the PBL height and the LCL (Figs. 29, 30), while the ShCu days in models result from a
much moister environment compared with that in the observations (Fig. 30). In general, the surface properties and fluxes are relatively better reproduced in the historical runs than in the AMIP runs (Fig. 28), with limited impact of resolutions.

Besides the NARRM configuration illustrated in the present study, E3SMv2 has been successfully run with RRM meshes with finer grids located at other regions (Antarctic, Arctic, and Southeast Pacific). We expect that the hybrid timestep strategy is a general approach that can be applied to these RRMs to simulate high-resolution climate at different areas. With that in mind,
we streamlined the process of creating new RRM configurations to facilitate broader RRM applications in the next phase of the E3SM project. Depending on the goal of RRM simulations, further improvements over the refined domain can be achieved via additional parameter tuning. In that case, nudging the outside coarser domain may be necessary to avoid severe degradations to the climate there. Such nudging capability is available in E3SM (e.g., Tang et al., 2019) and one has the option to nudge towards the data from reanalysis product or low-resolution E3SM simulation. Lastly, we highlight that this paper serves as an
overview of the NARRM atmosphere, land, and river models. More in-depth analysis is planned to be reported in follow-up papers.

*Code and data availability.* All E3SM codes may be accessed on GitHub at https://github.com/E3SM-Project/E3SM, including a mainte-nance branch (`maint-2.0`; https://github.com/E3SM-Project/E3SM/tree/maint-2.0) which has been created to reproduce these simulations.

Complete native model output as well as nudging simulations' climatology data are accessible directly at the National Energy Research
Scientific Computing Center (NERSC): https://portal.nersc.gov/archive/home/projects/e3sm/www/WaterCycle/E3SMv2/LR and https://portal.nersc.gov/archive/home/projects/e3sm/www/WaterCycle/E3SMv2/NARRM for low-resolution and NARRM simulations, which are documented at https://e3sm-project.github.io/e3sm_data_docs. A subset of the native output is also available through the DOE Earth System Grid



Federation (ESGF) at https://esgf-node.llnl.gov/search/e3sm/?model_version=2_0. Data reformatted following CMIP conventions will also be available through ESGF at https://esgf-node.llnl.gov/projects/e3sm.

Performance data and scripts are located at https://github.com/E3SM-Project/perf-data/tree/archive/v2-narrm-perf-study/v2-narrm.

## Appendix A

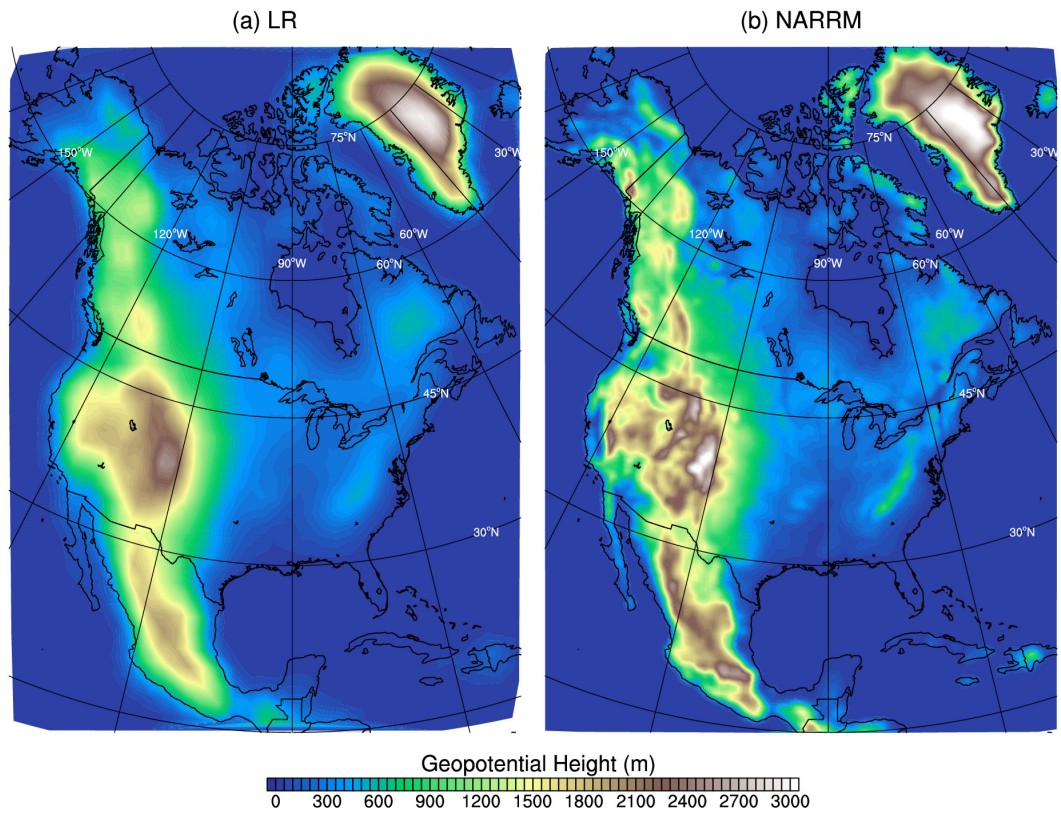

**Figure A1.** Surface geopotential height of (a) LR and (b) NARRM over North America.



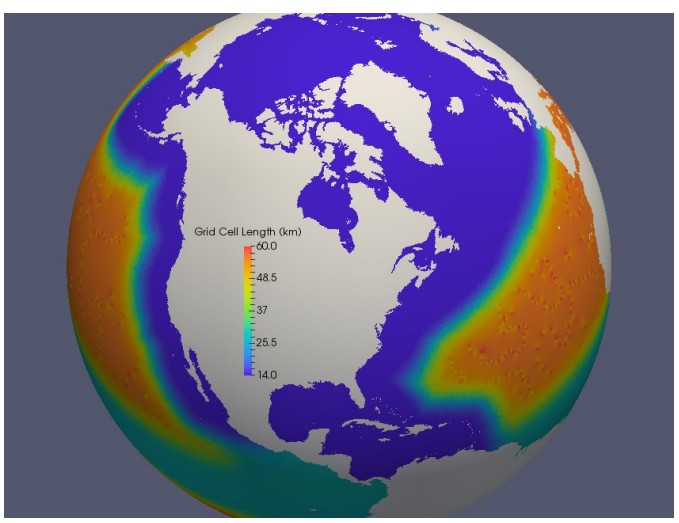

**Figure A2.** North American RRM (NARRM) grids for ocean and sea ice.

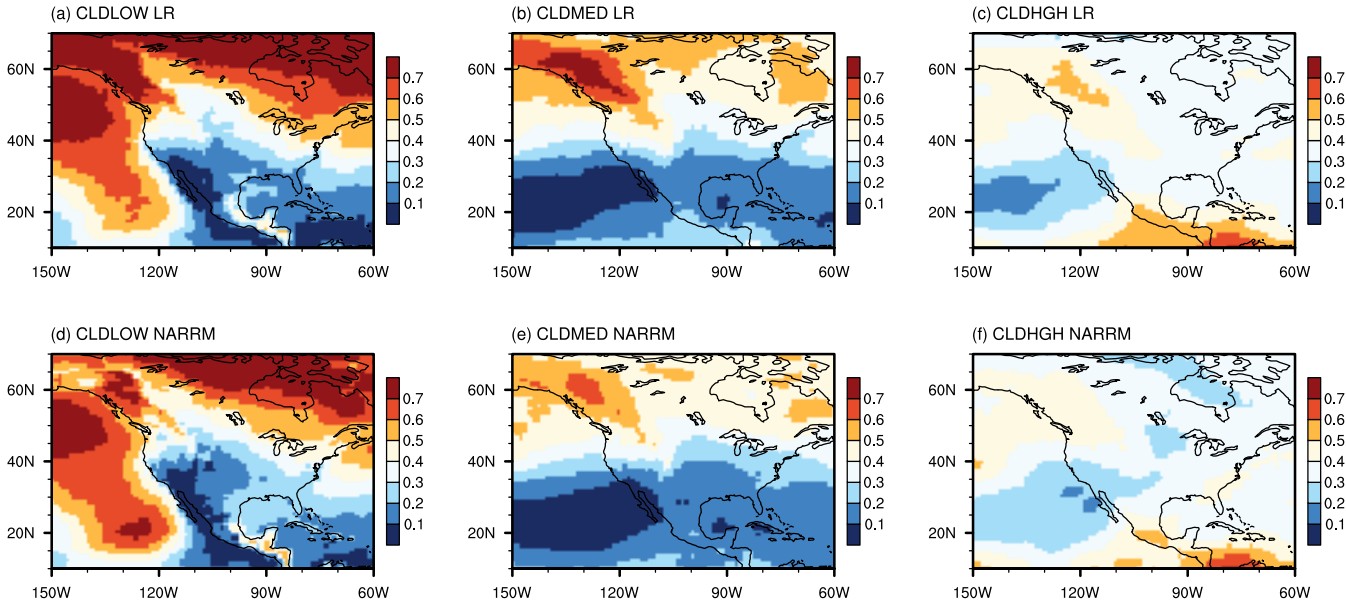

**Figure A3.** Cloud fractions over North America simulated by the nudged LR (top) and NARRM (bottom) for low clouds (CLDLOW; left), middle clouds (CLDMED; middel), and high clouds (CLDHGH; right).



**Table A1.** Definition criteria and sample size of the clear-sky regime (Clear) and shallow cumulus regime (ShCu) based on the ARM observations and four different model simulations.

| Regime | Definition criteria | Obs. | LR (A1-3) | LR (H1-5) | NARRM (A1-3) | NARRM (H1-5) |
|---|---|---|---|---|---|---|
| Clear | **Obs: Analysis period (2004-2012)**<br>• Precipitation rate = 0 mm/hr at all 24 hours<br>• Between 0800 and 1600 LST, total cloud fraction $\leq$ 15%, low-level and mid-level cloud fraction $\leq$ 5%, and high-level cloud fraction $\leq$ 10%<br><br>**Model: Analysis period (2004-2012)**<br>• Precipitation rate < 0.1 mm/hr at all 24 hours<br>• Between 0800 and 1600 LST, total cloud fraction $\leq$ 15%, low-level and mid-level cloud fraction $\leq$ 5%, and high-level cloud fraction $\leq$ 10% | 66 | 165 | 86 | 154 | 66 |
| ShCu | **Obs: Analysis period (2004-2012)**<br>• Precipitation rate = 0 mm/hr at all 24 hours<br>• Cloud tops < 4 km and cloud bases gradually rise with time over the day<br>• Above 4 km, there is usually no cloud or cloud fraction < 5%, except on a few days when there is some high cirrus above 10 km<br>• Satellite images of ShCu days identified based on Active Remote Sensing of Clouds data and the Total Sky Imager are examined manually to ensure that the cloud field develops homogeneously and is not affected by other large-scale weather phenomena<br><br>**Model: Analysis period (1980-2012)**<br>• Precipitation rate < 0.25 mm/hr at all 24 hours<br>• Diurnal maximum hourly low-level cloud fraction between 5% and 70%, and between 1000 and 1800 LST<br>• Between 0000 and 0600 LST, low-level cloud fraction < 5%<br>• Diurnal maximum hourly mid-level cloud fraction $\leq$ 10% and is lower than that of low-level cloud fraction at all 24 hours | 48 | 34 | 66 | 23 | 48 |



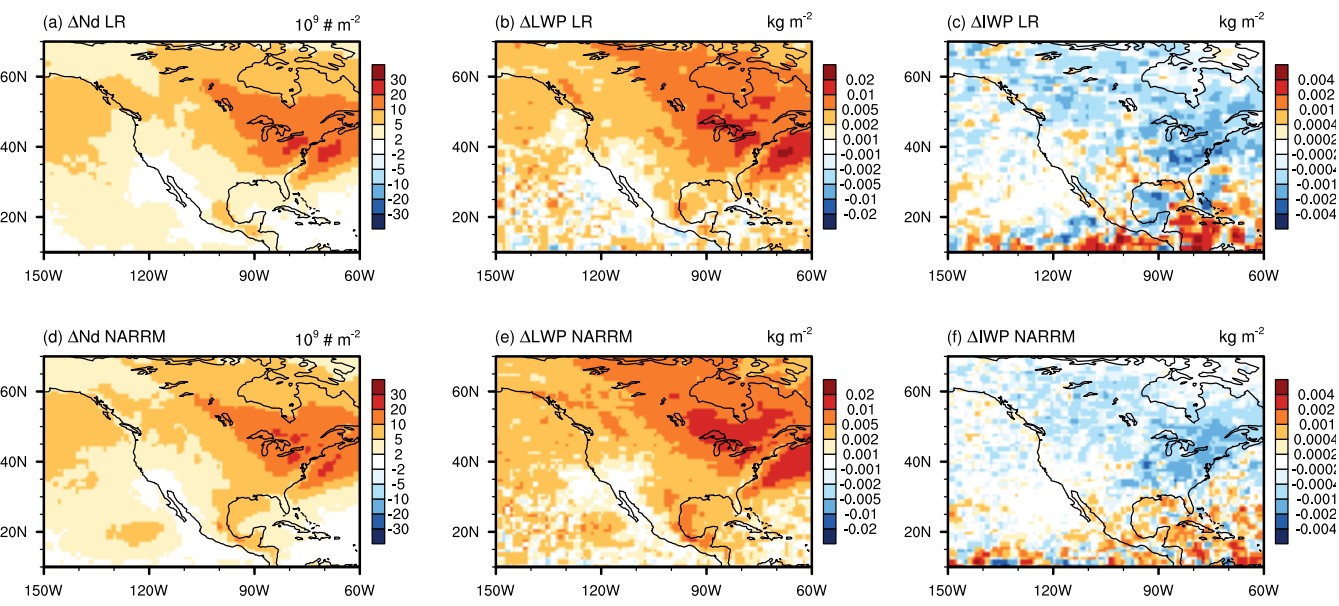

**Figure A4.** PD-PI (present-day (PD) minus pre-industrial (PI)) changes in cloud droplet number concentration (left), liquid water path (LWP), and ice water path (IWP) at North America calculated from the nudged LR (top) and NARRM (bottom) simulations.

*Author contributions.* As part of the E3SM water cycle group co-led by JCG and LPVR, QT led the Regionally Refined Model (RRM) development team with significant contributions from JCG, MAT, WL, BRH, PAU, and AMB, and help from OG, JDW, CZ, CSZ, ELR, AFR, MEM, RLJ, AVDV, PMC, and GB. The RRM configuration leverages the low-resolution model development effort from TZ, KZ, and XZ. QT and JCG designed the paper scope. QT, JCG, MAT, AMB, KZ, BS, and RMF carried out the model simulations with assistance from AM, NDK, and RLJ. QT led the analysis and the manuscript writing with significant contributions from JCG, MAT, WL, AMB, TZ, KZ, YZ, MZ, MW, HW, CT, AMR, YQ, HYL, and YF. All co-authors contributed to the manuscript.

*Competing interests.* At least one of the (co-)authors is a member of the editorial board of "Geoscientific Model Development".

*Acknowledgements.* This research was supported as part of the Energy Exascale Earth System Model (E3SM) project, funded by the U.S. Department of Energy, Office of Science, Office of Biological and Environmental Research (BER). E3SM production simulations were performed on a high-performance computing cluster provided by the BER Earth System Modeling program and operated by the Laboratory Computing Resource Center at Argonne National Laboratory, and the National Energy Research Scientific Computing Center (NERSC), a DOE Office of Science User Facility supported by the Office of Science of the U.S. Department of Energy under Contract No. DE-AC02-05CH11231. Developmental simulations were performed using BER Earth System Modeling program's Compy computing cluster located at Pacific Northwest National Laboratory and NERSC machine cori-knl. Additional developmental simulations, as well as post-processing and data archiving of production simulations used resources of NERSC.





Lawrence Livermore National Laboratory (LLNL) is operated by Lawrence Livermore National Security, LLC, for the U.S. Department of Energy, National Nuclear Security Administration under Contract DE-AC52-07NA27344. Support has been received from the LLNL LDRD projects 22-ERD-008, "Multiscale Wildfire Simulation Framework and Remote Sensing". Pacific Northwest National Laboratory is operated

by Battelle for the U.S. Department of Energy under Contract DE-AC05-76RL01830. This paper describes objective technical results and analysis. Any subjective views or opinions that might be expressed in the paper do not necessarily represent the views of the U.S. Department of Energy or the United States Government. Sandia National Laboratories is a multimission laboratory managed and operated by National Technology & Engineering Solutions of Sandia, LLC, a wholly owned subsidiary of Honeywell International Inc., for the U.S. Department of Energy's National Nuclear Security Administration under contract DE-NA0003525. This paper describes objective technical results and

analysis. Any subjective views or opinions that might be expressed in the paper do not necessarily represent the views of the U.S. Department of Energy or the United States Government.

This work was supported by the U.S. Department of Energy through the Los Alamos National Laboratory. Los Alamos National Laboratory is operated by Triad National Security, LLC, for the National Nuclear Security Administration of U.S. Department of Energy (Contract No. 89233218CNA000001).





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
