# Peer review of "The Fully Coupled Regionally Refined Model of E3SM Version 2: Overview of the Atmosphere, Land, and River Results"

_Geoscientific Model Development, 2022_

## Referee Comment (RC1)

Review of *The Fully Coupled Regionally Refined Model of E3SM Version 2: Overview of the Atmosphere, Land, and River*

Qi Tang and Coauthors

**General Impressions**

This manuscript provides the first documented regional refinement model (RRM) configuration fully coupled to an ocean model. Further, the ocean is regionally-refined as well, although the details are left to a paper currently in preparation. In accordance with the CMIP6 protocol, a suite of DECK experiments were carried out on a grid with 25 km refinement over North America (NARRM). Extensive analysis illustrates clears improvements in the climatology over the standard low resolution (LR) control. For this reason, I believe this work should be documented and communicated to the broader community.

However, I believe major revisions are required before this manuscript is in an acceptable form for publication. The paper is generally too long. It is clear that specialized subgroups were assigned a particular results section and the lead authors were responsible for stringing those sections together into a logical progression and coherent manuscript. I believe more needs to be done in this area. Here are some suggestions:

- Reduce non-results text in Introduction, Methods and Discussions and Conclusions section (generally referring to my a, b complaints below).

- Combine Section 5.1.5 on Cloud Feedback with Section 5.1.3 on North Polar Clouds (also Figure 12 seems like it should be in this combined section as well).

- Combine Section 5.2.1 on Hydrology and 5.4 River, as both sections contains runoff plots suggesting the analysis can be seamlessly combined.

Length and readability issues aside, I have two major concerns (a, b).

(a). Early on the authors develop "two criteria should be satisfied for the RRM to be widely adopted for global ESM releases: (1) reasonable global climate, and (2) minimal effort of retuning based upon the low-resolution counterpart." It's not clear what the authors mean in (2). They describe it as "Regarding (2), some physics parameterizations (e.g., deep convection) suffer from poor scale-awareness and hence require retuning as the model resolution increases (e.g., Xie et al., 2018). This implies significant model calibration efforts that may be unaffordable in addition to tuning the low-resolution model." This last sentence stating that tuning may be unaffordable does not constitute a "criteria" as the authors have chosen to characterize (2). This all depends on the resources available to the modeling institution to engage in an additional tuning effort for RRM — there is no requirement to use "minimal" resources as stated in (2).

Later on the authors state that their proposed solution to (2) "simplifies the RRM development as it naturally avoids further tuning the RRM beyond what was done for LR" (Line 147). This I think is the correct point that authors should flesh out. They simplified the process by using a hybrid time-step and therefore not requiring additional resources for tuning. They could have gone the hard way, full tuning, but they choose to use minimal resources.

The authors state that the usual practice of using a shorter physics time-step globally in RRM "cannot satisfy the two criteria above for the purpose of global climate production simulations mainly because the ZM deep convection scheme and other cloud parameterizations used by EAM are by design not scale-aware (Xie et al., 2018)." Here begins a pattern throughout the paper of conflating scale-awareness with time-step awareness in the cloud schemes. (2) is not a true criteria, and is more of a statement that it will take additional resources if you want to tune the physics at the shorter-physics time-step to produce a reasonable climate, i.e., criteria (1). The justification authors provide that they didn't tune because the physics aren't scale-aware does not make sense, as they are still running non scale-aware physics, just with a longer time-step. In many places throughout the paper they refer to how this hybrid time-step strategy is responsible for the success of satisfying both criteria. And then they also make statements like at line 678 where they say the hybrid time-step "bypasses the persistent poor scale-aware problem of atmospheric physics in a multi-scale framework," which is just not true.

(b). Hybrid time-step strategy. I have more details in the line by line comments, but my main issue here is the lack of discussion on the downsides of the hybrid time-step strategy. It is the resolved updrafts at higher-resolution that are responsible for the improvements in precipitation rates and extremes O'Brien et al. (2016). In GCMs, updrafts manifest through an instability between the physics (condensation and buoyancy) and the dynamics (vertical motion, adiabatic cooling and super-saturation). As the 25 km dynamics have larger vertical velocities and hence fast time-scales, it requires the physics time-step to be reduced in order to keep up with the the now faster evolving instability. These time-truncation errors due to the hybrid time-step approach can reduce vertical velocities by over half according to Appendix A in Herrington et al. (2019) or Figure 6 in Herrington and Reed (2018). Therefore by choosing the hybrid time-step strategy, one is choosing not to fully embrace the resolved updrafts that come with 25 km, and arguably not taking full advantage of all that 25 km can offer.

**Comments**

Line 69. "The RRM high-resolution results are robust regardless of where the fine-grid patch is located ..." I disagree with this statement based on the studies of Rauscher et al. (2013) (also the CAM4 simulations in the Zarzycki et al. (2014) paper you've cited). It's clear that putting the refinement region over the ITCZ can facilitate grid induced circulations that degrade the simulation quality over the low-res uniform resolution solution.

Line 86. Exactly what is meant by the (2) requirement "minimal effort of retuning based upon the low-resolution counterpart." Are the authors trying to convey that the tunings are based on the low resolution counterpart, and therefore we need to spend a "minimal"

amount of time retuning the RRM? Why shouldn't a modeling center spend just as much time tuning a RRM as their low resolution counterpart? I mean, the underlying issue is that it's expensive to tune the RRM, which I agree with, but that doesn't mean the modeling centers shouldn't try to do their due diligence by using a similar level of resources as used in the low resolution effort. Of course resources are limited, and it may not be practical to provide the same level of support to two versions (low-res and RRM). But if that's the case it's institution specific and needs to be stated as the justification for the authors design choices, not a blanket requirement for all modeling centers. I don't think it's out of the realm of possibilities that a modeling institution can rearrange their priorities to provide the same level of human support it provides to the low-resolution effort, plus the additional computational resources required to tune RRM.

Line 118. Change "with additional tuning" to "with additional modifications" as you cite enhancements, not just tunings.

Line 138. What does the lack of scale-awareness in the physics have to do with either of the two requirements? I suspect you mean that the lack of scale-awareness would require extensive tuning, and that's inconsistent with (2). As I've mentioned, I don't think (2) as it's written is an actual constraint.

Line 142. The claim that you can't get "optimal climate" even if you retuned the model because the physics aren't scale-aware. Have the authors tried? This is possible but it is not a foregone conclusion. The extent that a RRM solution is tainted by the lack of scale-aware physics depends on the region chosen for refinement, e.g., refining the ITCZ is more problematic (Rauscher et al. (2013)) than refining high-latitude regions where large amounts of buoyancy and convection are not being generated.

Line 144. This hybrid time-stepping strategy is not new (see Zarzycki et al. (2014) you've cited). While I agree with the benefits of this approach as the authors describe, what are you losing? You're increasing your truncation error substantially. See for example the truncation errors derived from moist bubble experiments in Appendix A of Herrington et al. (2019) (full citation below). A more more balanced discussion is needed here.

Line 187. Can you be more clear and explicitly state you are substituting out the smoothing algorithm in Lauritzen for an internal algorithm?

Figure 4. Could the authors add an outline of where the transition region is in this plot? It's a nice result, and I think that addition would better convey the result.

Line 250. What is the production i/o? Is there high-frequency output?

Figure 6. Are you using a fixed low-res reference for the dashed lines? Or are you changing the low-res reference for each processor count to be the equivalent low-res processor count?

Line 289. Maybe put the number of column in NARRM in parenthesis after NARRM, so the 14k number doesn't come out of nowhere?

Line 298. RMSE, not RSME.

Line 299. "for the first historical member ..." why not use the ensemble means in Figure 7?

Line 358. "highlights the advantage of coupled RRM over a single-component RRM ..." How can you conclude this? You are not comparing to RRM AMIP, you are comparing to LR coupled. In RRM AMIP, you can easily select to have a high-resolution coastline, and the ocean grid is nominally on the RRM grid.

Line 367. Is it correct to compare min/max values of field on different resolution grids?

Line 375. Can the authors provide more details on resolution dependent emission factors? Why is this a thing?

Line 381. "the same input datasets." Are the datasets coarser or finer than the RRM? If so, are the mapped emissions dataset aliased to their coarser grid?

Figure 16. I suspect this increase in AOD would survive remapping to a common grid before computing the histogram, but I still think this should be done to make it a more apples-to-apples comparison.

Line 537. "which is dominated by the reduced SW cloud feedback over the Northeastern US ... " Just a suggestion, but it would be useful if the cloud analysis in 5.1.3 could be extended beyond the North Polar Region so that we could compare the cloud feedback's in this section with existing cloud bias in the base state in LR and NARRM (and also extend the cloud feedbacks to the North Polar Region?).

Section 5.3. Why are the latent heat fluxes larger in LR(H1-5), and has systemically lower LCL on clear-sky days? Does NARRM introduce a dry bias in the atmosphere?

Figure 31. no corresponding a, b, c labels in the plot. What should be "a" is labeled as NARRM when it should be LR?

Figure A1. As topography is responsible for so many of the differences between LR and NARRM, I would suggest bringing this figure into the main paper.

**References**

A. Herrington and K. Reed. An idealized test of the response of the community atmosphere model to near-grid-scale forcing across hydrostatic resolutions. *J. Adv. Model. Earth Syst.*, 10(2):560–575, 2018.

A. R. Herrington, P. H. Lauritzen, K. A. Reed, S. Goldhaber, and B. E. Eaton. Exploring a lower resolution physics grid in cam-se-cslam. *Journal of Advances in Modeling Earth Systems*, 11, 2019.

T. A. O'Brien, W. D. Collins, K. Kashinath, O. Rübel, S. Byna, J. Gu, H. Krishnan, and P. A. Ullrich. Resolution dependence of precipitation statistical fidelity in hindcast simulations. *J. Adv. Model. Earth Syst.*, 8(2):976–990, 2016. doi: 10.1002/2016ms000671. URL http://dx.doi.org/10.1002/2016ms000671.

S. A. Rauscher, T. D. Ringler, W. C. Skamarock, and A. A. Mirin. Exploring a global multiresolution modeling approach using aquaplanet simulations. *Journal of Climate*, 26(8):2432–2452, 2013. doi: 10.1175/jcli-d-12-00154.1.

C. M. Zarzycki, M. N. Levy, C. Jablonowski, J. R. Overfelt, M. A. Taylor, and P. A. Ullrich. Aquaplanet experiments using cam's variable-resolution dynamical core. *J. Climate*, 27(14):5481–5503, 2014. doi: 10.1175/JCLI-D-14-00004.1.

---

## Referee Comment (RC2)

**Review of "The Fully Coupled Regionally Refined Model of E3SM Version 2: Overview of the Atmosphere, Land, and River" by Tang et al., submitted to** *Geoscientific Model Development*

**Recommendation:** Minor revision

**General comments:**

Many numerical studies have demonstrated the importance of fine resolution in modelling the climate system, but at the current time, fine resolution and long-term integrations are competing requirements for climate simulations due to limited computational resources. With these demands and limitations, the variable-resolution approach is an attractive method to conduct high-resolution simulations within a global lower-resolution model. In this study, the authors overview the E3SM version 2 with the regionally refined technique, and presented the evaluation of climate simulations. Results show that the regionally refined technique can provide improved simulations over regional scales. This manuscript would be suitable for publication after revision.

In section 4, model results are validated at the global scale in terms of global air temperature increase and climate sensitivities. Precipitation simulation skill is an important aspect. I suggest the authors to add a figure to show the comparison of global annual mean precipitation.

Besides, I found that although the variable-resolution method indeed improves many aspects in modeling the regional climate, but the systematic biases in the lower-resolution model are not reduced. The authors may give more explain about this point in the manuscript.

**Minor comments:**

Table 2. Need to indicate the meaning of the numbers in the brackets.

Figure 8. Need to indicate the observational minimum-maximum range in the legend.

Figure 11. In the ERA5 reanalysis, I think there are also uncertainties in the precipitation. Could you please add an observational benchmark?

---

## Author Comment (AC2)

**Response to Referee #1 (in blue)**

**General Impressions**

This manuscript provides the first documented regional refinement model (RRM) configuration fully coupled to an ocean model. Further, the ocean is regionally-refined as well, although the details are left to a paper currently in preparation. In accordance with the CMIP6 protocol, a suite of DECK experiments were carried out on a grid with 25 km refinement over North America (NARRM). Extensive analysis illustrates clears improvements in the climatology over the standard low resolution (LR) control. For this reason, I believe this work should be documented and communicated to the broader community.

However, I believe major revisions are required before this manuscript is in an acceptable form for publication. The paper is generally too long. It is clear that specialized sub-groups were assigned a particular results section and the lead authors were responsible for stringing those sections together into a logical progression and coherent manuscript. I believe more needs to be done in this area. Here are some suggestions:

Thank you for your positive assessment and constructive comments and suggestions, which certainly improve the quality of the manuscript. Indeed, that is the nature of this overview paper. The authors revised the paper following these suggestions.

- Reduce non-results text in Introduction, Methods and Discussions and Conclusions section (generally referring to my a, b complaints below).

  Thank you. Please see the details below about our revision.

- Combine Section 5.1.5 on Cloud Feedback with Section 5.1.3 on North Polar Clouds (also Figure 12 seems like it should be in this combined section as well).

  We agree and have combined these two sections as the reviewer suggested. In the revised manuscript, Section 5.1.3 describes the analysis of cloud and cloud feedback.

- Combine Section 5.2.1 on Hydrology and 5.4 River, as both sections contains runoff plots suggesting the analysis can be seamlessly combined.

  We combined these Sections as suggested. Section 5.2 on Land revised to three new subsections: 5.2.1 Snowpack, 5.2.2 Runoff and evapotranspiration, 5.2.3 Streamflow to improve readability.

Length and readability issues aside, I have two major concerns (a, b).

(a). Early on the authors develop "two criteria should be satisfied for the RRM to be widely adopted for global ESM releases: (1) reasonable global climate, and (2) minimal effort of retuning based upon the low-resolution counterpart." It's not clear what the authors mean in (2). They describe it as "Regarding (2), some physics parameterizations (e.g., deep convection) suffer from poor scale-awareness and hence require retuning as the model resolution increases (e.g., Xie et al., 2018). This implies significant model calibration efforts that may be unaffordable in addition to tuning the low-resolution model." This last sentence stating that tuning may be unaffordable does not constitute a

"criteria" as the authors have chosen to characterize (2). This all depends on the resources available to the modeling institution to engage in an additional tuning effort for RRM — there is no requirement to use "minimal" resources as stated in (2).

Later on the authors state that their proposed solution to (2) "simplifies the RRM development as it naturally avoids further tuning the RRM beyond what was done for LR" (Line 147). This I think is the correct point that authors should flesh out. They simplified the process by using a hybrid time-step and therefore not requiring additional resources for tuning. They could have gone the hard way, full tuning, but they choose to use minimal resources.

That's right. (2) is not a so necessary criteria as (1) for a global ESM release. We revised that paragraph in the introduction to have only one criteria: "At a minimum, the criteria of reasonable global climate should be satisfied for the RRM to be widely adopted for global ESM releases.".

(2) was removed from the requirements and was rephrased as an approach to simplify the RRM development as "We propose an innovative RRM strategy (see details in Section 2.1) to meet the criteria above with a minimal retuning effort and for the first time to deliver production climate simulations using a fully coupled RRM.".

The authors state that the usual practice of using a shorter physics time-step globally in RRM "cannot satisfy the two criteria above for the purpose of global climate production simulations mainly because the ZM deep convection scheme and other cloud parameterizations used by EAM are by design not scale-aware (Xie et al., 2018)." Here begins a pattern throughout the paper of conflating scale-awareness with time-step awareness in the cloud schemes. (2) is not a true criteria, and is more of a statement that it will take additional resources if you want to tune the physics at the shorter-physics time-step to produce a reasonable climate, i.e., criteria (1). The justification authors provide that they didn't tune because the physics aren't scale-aware does not make sense, as they are still running non scale-aware physics, just with a longer time-step. In many places throughout the paper they refer to how this hybrid time-step strategy is responsible for the success of satisfying both criteria. And then they also make statements like at line 678 where they say the hybrid time-step "bypasses the persistent poor scale-aware problem of atmospheric physics in a multi-scale framework," which is just not true.

We agree with the reviewer that (2) should not be a criterion and have revised the manuscript accordingly (see the answer to the question above).

Furthermore, we pointed out in the second to the last paragraph of introduction that based on our EAMv1 development experience even if we had the resources to full tune RRM like we did for EAMv1, "the better RRM global climate is not warranted by retuning based on our EAMv1 RRM experience".

For v1, we have tuned models at both low-resolution (100 km) (Golaz et al., 2019) and high-resolution (25 km) (Caldwell et al., 2019). The figure below shows the global spatial root-mean-square-error (RMSE) results from the v1.LR (blue triangles) and the v1.RRM (red triangles) with high-resolution grids over the CONUS and both physics parameters and

timesteps tuned for high-resolution (see details in Tang et al., 2019). It is clear that the v1.RRM results are generally worse (larger RMSE) than the v1.LR results. This is the main reason why we didn't choose this approach for the NARRM in E3SMv2.

We included this v1 figure as Fig. A2 in the revised version and clarified this in Section 2.1. It now reads as "Furthermore, even if NARRM used the retuned high-resolution physics parameters along with the shorter physics timestep, we would still have degraded global simulation quality over the LR model (see Fig. A2 for the EAMv1 results).".

In addition, we revised the sentence "bypasses the persistent poor scale-aware problem of atmospheric physics in a multi-scale framework" to "mitigates the negative impacts caused by the persistent poor scale-aware problem of atmospheric physics in a multi-scale framework".

[Figure]

(b). Hybrid time-step strategy. I have more details in the line by line comments, but my main issue here is the lack of discussion on the downsides of the hybrid time-step strategy. It is the resolved updrafts at higher-resolution that are responsible for the improvements in precipitation rates and extremes O'Brien et al. (2016). In GCMs, updrafts manifest through an instability between the physics (condensation and buoyancy) and the dynamics (vertical motion, adiabatic cooling and super-saturation). As the 25 km dynamics have larger vertical velocities and hence fast time-scales, it requires the physics time-step to be reduced in order to keep up with the now faster

evolving instability. These time-truncation errors due to the hybrid time-step approach can reduce vertical velocities by over half according to Appendix A in Herrington et al. (2019) or Figure 6 in Herrington and Reed (2018). Therefore by choosing the hybrid time-step strategy, one is choosing not to fully embrace the resolved updrafts that come with 25 km, and arguably not taking full advantage of all that 25 km can offer.

That is right. Although the downside of the hybrid timestep strategy was implied, we didn't discuss it explicitly. We added the following sentences at the end of Section 2.1 to note that this strategy is more of a practical choice and balance the discussion by pointing out the disadvantage.

"It is worthwhile noting that the hybrid timestep strategy is a practical choice before the scale-aware cloud parameterization becomes available. With the coarsened physics timestep, NARRM cannot take full advantage of resolved physics (e.g., updrafts) at 25 km."

**Comments**

Line 69. "The RRM high-resolution results are robust regardless of where the fine-grid patch is located ..." I disagree with this statement based on the studies of Rauscher et al. (2013) (also the CAM4 simulations in the Zarzycki et al. (2014) paper you've cited). It's clear that putting the refinement region over the ITCZ can facilitate grid induced circulations that degrade the simulation quality over the low-res uniform resolution solution.

Thanks. We missed that RRM over ITCZ degrades the model simulation. We tuned down the statement and it reads as "The RRM high-resolution results are robust for most places except the Inter-Tropical Convergence Zone (Rauscher et al., 2013; Zarzycki et al., 2014)…".

Line 86. Exactly what is meant by the (2) requirement "minimal effort of retuning based upon the low-resolution counterpart." Are the authors trying to convey that the tunings are based on the low resolution counterpart, and therefore we need to spend a "minimal"

amount of time retuning the RRM? Why shouldn't a modeling center spend just as much time tuning a RRM as their low resolution counterpart? I mean, the underlying issue is that it's expensive to tune the RRM, which I agree with, but that doesn't mean the modeling centers shouldn't try to do their due diligence by using a similar level of resources as used in the low resolution effort. Of course resources are limited, and it may not be practical to provide the same level of support to two versions (low-res and RRM). But if that's the case it's institution specific and needs to be stated as the justification for the authors design choices, not a blanket requirement for all modeling centers. I don't think it's out of the realm of possibilities that a modeling institution can rearrange their priorities to provide the same level of human support it provides to the low-resolution effort, plus the additional computational resources required to tune RRM.

We agree that (2) is not a necessary requirement and removed it in the revised version. Details were provided in the responses to the concern (a) above.

Line 118. Change "with additional tuning" to "with additional modifications" as you cite enhancements, not just tunings.

Good point. Done. Thanks.

Line 138. What does the lack of scale-awareness in the physics have to do with either of the two requirements? I suspect you mean that the lack of scale-awareness would require extensive tuning, and that's inconsistent with (2). As I've mentioned, I don't think (2) as it's written is an actual constraint.

We showed in the answer above that the EAMv1 CONUS RRM with high-resolution timesteps and physics tuning simulates worse global climate than the low-resolution model. The revised text reads as "Such treatment (at least for the EAMv1 CONUS RRM) degrades the global simulation quality over the low-resolution model and thus is not suitable for global climate production simulations."

Line 142. The claim that you can't get "optimal climate" even if you retuned the model because the physics aren't scale-aware. Have the authors tried? This is possible but it is not a foregone conclusion. The extent that a RRM solution is tainted by the lack of scale-aware physics depends on the region chosen for refinement, e.g., refining the ITCZ is more problematic (Rauscher et al. (2013)) than refining high-latitude regions where large amounts of buoyancy and convection are not being generated.

Yes, we tried to retune the EAMv1 CONUS RRM and could not get optimal climate (see the results in the answer to the concern (a)).

Line 144. This hybrid time-stepping strategy is not new (see Zarzycki et al. (2014) you've cited). While I agree with the benefits of this approach as the authors describe, what are you losing? You're increasing your truncation error substantially. See for example the truncation errors derived from moist bubble experiments in Appendix A of Herrington et al. (2019) (full citation below). A more more balanced discussion is needed here.

Thanks for the suggestion – A more balanced discussion is helpful to understand the pros and cons of the hybrid timestep approach. Please see in the answers to the concern (b) above where we revised the manuscript to point out "NARRM cannot take full advantage of resolved physics (e.g., updrafts) at 25 km".

Line 187. Can you be more clear and explicitly state you are substituting out the smoothing algorithm in Lauritzen for an internal algorithm?

These two sentences were edited to make this point more clear. They now read "To generate topography and associated surface roughness fields on the NARRM grid, we rely on the tool chain described in Lauritzen et al. (2015) combined with a topography smoothing tool included with HOMME. The use of HOMME's topography smoothing tool ensures that the smoothing is done with the same discrete Laplace operator used internally in the dynamical core."

Figure 4. Could the authors add an outline of where the transition region is in this plot? It's a nice result, and I think that addition would better convey the result.

Agreed, this does make for a better plot. We changed the figure to show contour lines of temperature overlayed on a map of resolution denoted by a color scale (using the same scale as in Figure 2). It would have been nicer to show the grid directly on the plot, but in the high-resolution region the grid is too fine to plot the individual elements.

Line 250. What is the production i/o? Is there high-frequency output?

The settings of production I/O are pasted below for atmosphere (user_nl_eam), land (user_nl_elm), and river (user_nl_mosart).

High-frequency output are included in all three components where the *nhtfrq (output frequency) options are "-24" (daily), "-6" (6-hourly), or "-3" (3-hourly).

```
cat << EOF >> user_nl_eam
 nhtfrq =   0,-24,-6,-6,-3,-24,0
 mfilt  = 1,30,120,120,240,30,1
 avgflag_pertape = 'A','A','I','A','A','A','I'
 fexcl1  = 'CFAD_SR532_CAL', 'LINOZ_DO3', 'LINOZ_DO3_PSC', 'LINOZ_O3CLIM',
'LINOZ_O3COL', 'LINOZ_SSO3', 'hstobie_linoz'
 fincl1   = 'extinct_sw_inp','extinct_lw_bnd7','extinct_lw_inp','CLD_CAL', 'TREFMNAV',
'TREFMXAV'
 fincl2                                                                            =
'FLUT','PRECT','U200','V200','U850','V850','Z500','OMEGA500','UBOT','VBOT','TREFHT','
TREFHTMN:M','TREFHTMX:X','QREFHT','TS','PS','TMQ','TUQ','TVQ','TOZ',      'FLDS',
'FLNS', 'FSDS', 'FSNS', 'SHFLX', 'LHFLX', 'TGCLDCWP', 'TGCLDIWP', 'TGCLDLWP',
'CLDTOT', 'T250', 'T200', 'T150', 'T100', 'T050', 'T025', 'T010', 'T005', 'T002', 'T001', 'TTOP',
'U250', 'U150', 'U100', 'U050', 'U025', 'U010', 'U005', 'U002', 'U001', 'UTOP', 'FSNT', 'FLNT'
 fincl3 = 'PSL','T200','T500','U850','V850','UBOT','VBOT','TREFHT', 'Z700', 'TBOT:M'
 fincl4 = 'FLUT','U200','U850','PRECT','OMEGA500'
 fincl5 = 'PRECT','PRECC','TUQ','TVQ','QFLX','SHFLX','U90M','V90M'
 fincl6                                                                            =
'CLDTOT_ISCCP','MEANCLDALB_ISCCP','MEANTAU_ISCCP','MEANPTOP_ISCCP','M
EANTB_ISCCP','CLDTOT_CAL','CLDTOT_CAL_LIQ','CLDTOT_CAL_ICE','CLDTOT_C
AL_UN','CLDHGH_CAL','CLDHGH_CAL_LIQ','CLDHGH_CAL_ICE','CLDHGH_CAL_U
N','CLDMED_CAL','CLDMED_CAL_LIQ','CLDMED_CAL_ICE','CLDMED_CAL_UN','C
LDLOW_CAL','CLDLOW_CAL_LIQ','CLDLOW_CAL_ICE','CLDLOW_CAL_UN'
 fincl7 = 'O3', 'PS', 'TROP_P'
EOF
```

```
cat << EOF >> user_nl_elm
 hist_dov2xy = .true.,.true.
 hist_fincl2 = 'H2OSNO', 'FSNO', 'QRUNOFF', 'QSNOMELT', 'FSNO_EFF', 'SNORDSL',
'SNOW', 'FSDS', 'FSR', 'FLDS', 'FIRE', 'FIRA'
 hist_mfilt = 1,365
 hist_nhtfrq = 0,-24
 hist_avgflag_pertape = 'A','A'
EOF

cat << EOF >> user_nl_mosart
 rtmhist_fincl2 = 'RIVER_DISCHARGE_OVER_LAND_LIQ'
 rtmhist_mfilt = 1,365
 rtmhist_ndens = 2
 rtmhist_nhtfrq = 0,-24
EOF
```

Figure 6. Are you using a fixed low-res reference for the dashed lines? Or are you changing the low-res reference for each processor count to be the equivalent low-res processor count?

We are using one fixed low-res reference for each dashed line and not changing the reference for each processor count. To clarify this point, we have added the following sentence: "The single LR XS-layout ocean throughput value is used as the reference for the ocean and, similarly, the single LR XS-layout atmosphere throughput value is used as the reference for the atmosphere; these are the only measured data inputs to the performance models."

Line 289. Maybe put the number of column in NARRM in parenthesis after NARRM, so the 14k number doesn't come out of nowhere?

Thanks. We have clarified the role of number of elements in the calculation as follows: "For example, a uniform high-resolution atmosphere model would use the ne120pg2 grid, which has $6 \cdot 120^2$ elements. Using the same time steps and number of vertical levels as in the NARRM configuration, which has 14454 elements, for fixed computational resources, the high-resolution atmosphere configuration's throughput would be $6 \cdot 120^2 / 14454 = 5.98$ times smaller than the NARRM configuration's throughput, where this factor is the quotient of the numbers of elements in each of the two grids."

Line 298.  RMSE, not RSME.

Good catch - Changed. Thanks.

Line 299.  "for the first historical member ..." why not use the ensemble means in Figure 7?

Figure 7 can be replotted with ensemble means, but cannot be easily done for CMIP6 models, which have different ensemble sizes.  Whether using the first historical member or the ensemble mean should not change the Figure 7 conclusion on the global climatology comparison, which is normally robust with a single member, and we confirmed with the E3SM results (not shown).

Line 358. "highlights the advantage of coupled RRM over a single-component RRM ..." How can you conclude this?  You are not comparing to RRM AMIP, you are comparing to LR coupled. In RRM AMIP, you can easily select to have a high-resolution coastline, and the ocean grid is nominally on the RRM grid.

We agree that the evidence for benefits of the coupled RRM is not clear from just looking at fully coupled low resolution and fully coupled NARRM. A deeper analysis of the benefit of the fully coupled RRM is conducted in a companion paper nearing submission by Van Roekel et al. To clarify this point we have added one new figure, which is reproduced here.

[Figure]

All figures show the difference between modeled SST and the HadISST dataset. Panel (a) is the fully coupled RRM, (b) is the fully coupled LR simulation, (c) couples the RRM atmosphere to the LR ocean, and (d) the LR atmosphere couples to the RRM ocean.

In the stratus region near California (red box), we see that the SST bias is greatly reduced in the coupled RRM relative to the coupled LR.  Panels (c) and (d) show that the high-resolution atmosphere is primarily responsible for the reduction in bias.  However, if the bias in panel (c) is compared to (a) the fully coupled RRM further reduces the bias, suggesting resolution improvements in both components improve the coastal SST bias.

Line 367.  Is it correct to compare min/max values of field on different resolution grids?

We added panel (c) (old (c) becomes (d)) for the NARRM results on the same 100-km grids as

the LR results. The NARRM maximum value decreases from 6.71 mm/day to 5.61 mm/day after coarsening the resolution but remains greater than the LR maximum of 3.10 mm/day. So, the conclusion remains unchanged. We revised the text as "NARRM reduces the underestimation of maximum diurnal cycle magnitude with an 80% greater value (5.61 mm/day vs.~3.10 mm/day) than LR on the same 100-km grids.  The NARRM maximum can be as high as 6.71 mm/day on the 25-km grids, but still biases low compared to the observation (10.89 mm/day).".

Line 375. Can the authors provide more details on resolution dependent emission factors? Why is this a thing?

Yes, this is because the calculation of these natural aerosol emission fluxes depends on the model resolved small-scale surface winds, e.g., as shown in Feng et al. (2022), "without tuning the dust emission parameters, increasing the model horizontal resolution by a factor of 4 from the E3SMv1 low resolution (~110 km) to the higher resolution (~25 km) simulation results in about a 29% increase of global dust emission fluxes from 4,702 to 6,044 Tg per yr."

To clarify, we revised the sentence as below:

"since NARRM employs the hybrid timestep approach that eliminates re-tuning the scale-dependent aerosol parameters used in LR, e.g.,  the global scaling factor used to constrain the total emission fluxes of natural aerosols (dust and sea salt) which depend nonlinearly on the model-resolved small-scale surface winds, we will also discuss the impact of increasing model horizontal resolution on the natural aerosols and total aerosol optical depth (AOD) in NARRM."

Line 381. "the same input datasets." Are the datasets coarser or finer than the RRM? If so, are the mapped emissions dataset aliased to their coarser grid?

The anthropogenic aerosol and fire emissions data have a coarser resolution (1.9 degree latitude x 2.5 degree longitude) than the RRM grids. These emission datasets are bilinearly interpreted online to the grids the model runs on. Yes, the mapped emissions datasets are aliased to their coarser grids.

Figure 16.  I suspect this increase in AOD would survive remapping to a common grid before computing the histogram, but I still think this should be done to make it a more apples-to-apples comparison.

Figure 16 (now Fig. 18 in the revised manuscript) indeed compares the two model simulations (LR and NARRM) of dust AOD on a common grid: 0.25x0.25. This is now clarified in the text:

"Figure 18 shows an example of the calculated probability density function (PDF) for dust AOD over the major dust source region in the US (32-42 deg N, 118-108 deg W; indicated by the 10 deg x10 deg box in Fig. 15b), from both the LR (0101) and NARRM (0101) simulations in 2000-2014, which are remapped to the same 0.25 deg x 0.25 deg grid resolution. It is worth noting that the remapping of the LR results to the finer resolution leads to little improvement in the resolved spatial variability in dust AOD. Clearly, NARRM predicts more occurrences of high dust AOD over this region than LR, e.g., 22% of the dust AODs predicted by NARRM exceeds 0.015, which is the top 98th percentile of the LR model

predictions remapped to the same resolution."

Also clarified in the Figure 18 caption: "Calculated probability density function (PDF) of the dust AOD predictions from LR (0101) and NARRM (0101) between 2000--2014 remapped to the same 0.25 deg grid resolution, over the major dust source region in US …"

For completeness, we plot the dust AOD pdfs from LR and NARRM in their original resolutions below, which is similar to the original Fig. 16 (now Fig. 18 in the revised manuscript):

[Figure]

Line 537. "which is dominated by the reduced SW cloud feedback over the Northeastern US ... " Just a suggestion, but it would be useful if the cloud analysis in 5.1.3 could be extended beyond the North Polar Region so that we could compare the cloud feedback's in this section with existing cloud bias in the base state in LR and NARRM (and also extend the cloud feedbacks to the North Polar Region?).

In the revised manuscript, we combined the North polar region section and cloud feedback section into cloud and cloud feedback section following reviewer's suggestion. By doing so, we extended both the cloud and cloud feedback analysis to the same region that covers the entire NA.

Section 5.3. Why are the latent heat fluxes larger in LR(H1-5), and has systemically lower LCL on clear-sky days? Does NARRM introduce a dry bias in the atmosphere?

To explore why the latent heat fluxes are larger in LR (H1-5), here, we examined the summertime diurnal cycle of precipitation (shown below). During the daytime (0900-1800 LST), the precipitation distribution is similar across all four versions of model simulations. However, the nighttime precipitation is better in LR (H1-5) as compared with ARM observation, which is mainly attributed to the increased convective precipitation. The early morning RH profile also confirms a much moister lower and upper atmosphere in LR (H1-5). As a result, the LCL is much lower in LR (H1-5) compared to other model versions. The SST may also play a role for the performance of the coupled run with interactive ocean/ice, i.e., LR (H1-5). Further analysis is needed to fully resolve this question, which is beyond the scope of this study.

The warm and dry bias over the central U.S. is one of the common issues in weather and climate models. As shown in the mixing diagram (Fig. 32 in the revised manuscript), the surface is still warm and dry for all models on clear-sky days. NARRM does not introduce an additional dry bias in the atmosphere. It seems the impact of resolution is generally minor on the land-atmosphere coupling based on this study, however, we still need to look into the nighttime precipitation performance in LR to resolve the puzzle.

[Figure]

Figure 31. no corresponding a, b, c labels in the plot. What should be "a" is labeled as NARRM when it should be LR?

Done. Thanks.

Figure A1. As topography is responsible for so many of the differences between LR and NARRM, I would suggest bringing this figure into the main paper.

Agreed. We moved Figure A1 to the main text as Figure 8.

**References**

A. Herrington and K. Reed. An idealized test of the response of the community atmosphere model to near-grid-scale forcing across hydrostatic resolutions. *J. Adv. Model. Earth Syst.*, 10(2):560–575, 2018.

A. R. Herrington, P. H. Lauritzen, K. A. Reed, S. Goldhaber, and B. E. Eaton. Exploring a lower resolution physics grid in cam-se-cslam. *Journal of Advances in Modeling Earth Systems*, 11, 2019.

T. A. O'Brien, W. D. Collins, K. Kashinath, O. Rübel, S. Byna, J. Gu, H. Krishnan, and P. A. Ullrich. Resolution dependence of precipitation statistical fidelity in hindcast simulations. *J. Adv. Model.*

*Earth Syst.*, 8(2):976–990, 2016. doi: 10.1002/2016ms000671. URL `http://dx.doi.org/10.1002/2016ms000671`.

S. A. Rauscher, T. D. Ringler, W. C. Skamarock, and A. A. Mirin. Exploring a global multiresolution modeling approach using aquaplanet simulations. *Journal of Climate*, 26(8):2432–2452, 2013. doi: 10.1175/jcli-d-12-00154.1.

C. M. Zarzycki, M. N. Levy, C. Jablonowski, J. R. Overfelt, M. A. Taylor, and P. A. Ullrich. Aquaplanet experiments using cam's variable-resolution dynamical core. *J. Climate*, 27(14):5481–5503, 2014. doi: 10.1175/JCLI-D-14-00004.1.

Golaz, J.-C., Caldwell, P. M., Van Roekel, L. P., Petersen, M. R., Tang, Q., Wolfe, J. D., et al. (2019). The DOE E3SM coupled model version 1: Overview and evaluation at standard resolution. Journal of Advances in Modeling Earth Systems, 11, 2089–2129. https://doi. org/10.1029/2018MS001603

Caldwell, P. M., Mametjanov, A., Tang, Q., Van Roekel, L. P., Golaz, J.-C., Lin, W. et al. (2019). The DOE E3SM coupled model version 1: Description and results at high resolution. Journal of Advances in Modeling Earth Systems, 11, 4095–4146. https://doi.org/10.1029/2019MS001870

Tang, Q., Klein, S. A., Xie, S., Lin, W., Golaz, J.-C., Roesler, E. L., Taylor, M. A., Rasch, P. J., Bader, D. C., Berg, L. K., Caldwell, P., Giangrande, S. E., Neale, R. B., Qian, Y., Riihimaki, L. D., Zender, C. S., Zhang, Y., and Zheng, X.: Regionally refined test bed in E3SM atmosphere model version 1 (EAMv1) and applications for high-resolution modeling, Geosci. Model Dev., 12, 2679–2706, https://doi.org/10.5194/gmd-12-2679-2019, 2019.

---

## Author Comment (AC3)

**Response to Referee #2 (in blue)**

**Recommendation:** Minor revision

**General comments:**

Many numerical studies have demonstrated the importance of fine resolution in modelling the climate system, but at the current time, fine resolution and long-term integrations are competing requirements for climate simulations due to limited computational resources. With these demands and limitations, the variable-resolution approach is an attractive method to conduct high-resolution simulations within a global lower-resolution model. In this study, the authors overview the E3SM version 2 with the regionally refined technique, and presented the evaluation of climate simulations. Results show that the regionally refined technique can provide improved simulations over regional scales. This manuscript would be suitable for publication after revision.

Thank you, and that is our main purpose to document the fully coupled E3SMv2 NARRM results.

In section 4, model results are validated at the global scale in terms of global air temperature increase and climate sensitivities. Precipitation simulation skill is an important aspect. I suggest the authors to add a figure to show the comparison of global annual mean precipitation.

Thanks for the suggestion. We added in Section 4 the description of the global annual mean precipitation results (Fig. A3), which show that NARRM and LR have very similar global patterns.

Besides, I found that although the variable-resolution method indeed improves many aspects in modeling the regional climate, but the systematic biases in the lower-resolution model are not reduced. The authors may give more explain about this point in the manuscript.

As much as we would love to, NARRM is not expected to reduce the systematic biases compared to the LR model as the same low-resolution grids still cover most of the global area in NARRM.

**Minor comments:**

Table 2. Need to indicate the meaning of the numbers in the brackets.

The numbers in the brackets indicate the simulation year numbers. We revised the table caption to clarify this. Thanks.

Figure 8. Need to indicate the observational minimum-maximum range in the legend.

Revised the figure legend to show the minimum and maximum numbers of observations.

Figure 11. In the ERA5 reanalysis, I think there are also uncertainties in the precipitation. Could you please add an observational benchmark?

Figure 11 mainly shows that the precipitation patterns are better captured by NARRM relative to LR in mountain areas. While we acknowledge that uncertainties exist in the ERA5 precipitation data, such uncertainties should not change this main point and hence we think ERA5 is a reasonable reference in this context.

---

## Author Response (AR2)

**Responses to editor comments** (responses are in blue)

Thank you for your revisions; you seem to have satisfied most of the reviewers' concerns. There are just a couple more:

Response: Thanks for the suggestions, the manuscript was revised accordingly.

(1) This sentence is unclear: "Furthermore, the better RRM global climate is not warranted by retuning based on our EAMv1 RRM experience." I think it'd be better rewritten as, "Furthermore, based on our EAMv1 RRM experience, retuning does not guarantee improved global climate performance." Does that fit with the intended meaning?

Response: Yes, that is exactly what we meant. We adopted the revised sentence. Thanks.

(2) You discuss a number of advantages of the hybrid time-step strategy. Are there also any disadvantages? (Presumably there are, or else it would have always been used, right?) If so, please add some discussion of these.

Response: That is right. There are disadvantages of the hybrid time-step strategy, which are due to the poor scale-aware deep convection scheme and other cloud parameterizations. Although these disadvantages are not inherent to the RRM and will go away when scale-awared schemes are available, which are active research topics, they impose negative impacts on the RRM climate performance.

We added a more balanced discussion about the hybrid time-step strategy. The last paragraph of Section 2.1 now reads as the following:
"It is worthwhile noting that the hybrid timestep strategy is a practical choice before the scale-aware cloud parameterization becomes available. With the coarsened physics timestep, NARRM cannot take full advantage of resolved processes (e.g., updrafts) at 25 km because the dynamics at 25 km explicitly resolve greater vertical velocities relative to those at 100 km and hence have faster dynamical time-scales, which require the correspondingly shortened physics timestep to match the faster evolving instability. The time-truncation errors of the hybrid timestep method are large at 25 km as quantified by a moist bubble test (Herrington et al., 2019)."